# Associations of cardiovascular biomarkers and plasma albumin with exceptional survival to the highest ages

Takumi Hirata[1,2], Yasumichi Arai [1,3✉], Shinsuke Yuasa [4], Yukiko Abe[1], Michiyo Takayama [5], Takashi Sasaki [1], Akira Kunitomi[4], Hiroki Inagaki [6], Motoyoshi Endo[7,8], Jun Morinaga[7], Kimio Yoshimura[9], Tetsuo Adachi[10], Yuichi Oike[7], Toru Takebayashi[11], Hideyuki Okano [1,3,12] & Nobuyoshi Hirose[1]

Supercentenarians (those aged ≥110 years) are approaching the current human longevity limit by preventing or surviving major illness. Identifying specific biomarkers conducive to exceptional survival might provide insights into counter-regulatory mechanisms against aging-related disease. Here, we report associations between cardiovascular disease-related biomarkers and survival to the highest ages using a unique dataset of 1,427 oldest individuals from three longitudinal cohort studies, including 36 supercentenarians, 572 semi-supercentenarians (105–109 years), 288 centenarians (100–104 years), and 531 very old people (85–99 years). During follow-up, 1,000 participants (70.1%) died. Overall, N-terminal pro-B-type natriuretic peptide (NT-proBNP), interleukin-6, cystatin C and cholinesterase are associated with all-cause mortality independent of traditional cardiovascular risk factors and plasma albumin. Of these, low NT-proBNP levels are statistically associated with a survival advantage to supercentenarian age. Only low albumin is associated with high mortality across age groups. These findings expand our knowledge on the biology of human longevity.

[1] Centre for Supercentenarian Medical Research, Keio University School of Medicine, Tokyo, Japan. [2] Department of Public Health, Hokkaido University Faculty of Medicine, Sapporo, Japan. [3] Keio University Global Research Institute, Tokyo, Japan. [4] Department of Cardiology, Keio University School of Medicine, Tokyo, Japan. [5] Centre for Preventive Medicine, Keio University School of Medicine, Tokyo, Japan. [6] Tokyo Metropolitan Institute of Gerontology, Tokyo, Japan. [7] Department of Molecular Genetics, Graduate School of Medical Sciences, Kumamoto University, Kumamoto, Japan. [8] Department of Molecular Biology, University of Occupational and Environmental Health, Fukuoka, Japan. [9] Department of Health Policy and Management, Keio University School of Medicine, Tokyo, Japan. [10] Department of Biomedical Pharmaceutics, Laboratory of Clinical Pharmaceutics, Gifu Pharmaceutical University, Gifu, Japan. [11] Department of Preventive Medicine and Public Health, Keio University School of Medicine, Tokyo, Japan. [12] Department of Physiology, Keio University School of Medicine, Tokyo, Japan. ✉email: yasumich@keio.jp

Aging is a dominant risk factor for most fatal diseases, such as cardiovascular disease, type 2 diabetes mellitus, Alzheimer's disease, and cancers[1,2]. Of these, cardiovascular disease is conspicuous as a leading cause of death in older adults[3]. Globally, deaths from cardiovascular disease increased by 40.8% between 1990 and 2013, climbing from 12.3 million to 17.3 million deaths; the aging population is a major driver of this trend in cardiovascular mortality[4]. Improved therapeutic interventions are needed to reduce age-related cardiovascular morbidity and mortality, especially in the face of a heart failure pandemic[5]. Understanding the molecular effects of aging on the cardiovascular system and the exploitation of endogenous counter-regulatory mechanisms are essential to the development of such therapeutic interventions.

Despite their extreme old age, centenarians are at low cardiometabolic risk, with preferable lipid profiles[6], a low prevalence of type 2 diabetes, and preserved insulin sensitivity[7] which result in delayed, or even ameliorated, cardiovascular morbidity[8]. However, recent electrocardiography (ECG) studies have reported that a substantial subpopulation of centenarians exhibits ECG abnormalities, such as left ventricular hypertrophy, abnormal Q waves, or atrial fibrillation, which are associated with increased mortality beyond 100 years of age[9,10]. These findings raise the fundamental question of whether clinical and subclinical cardiovascular disease is an inevitable consequence of extended longevity, even in the absence of modifiable risk factors.

Owing to their extremely low prevalence, supercentenarians (i.e., individuals who have reached their 110th birthday) have only recently emerged as a subject for descriptive study. According to the 1900 cohort life tables in Japan, 122 men and 617 women per 100,000 births reached 100 years of age in 2000. Of those, 0.9% (0.3 men and 6.6 women per 100,000 births) reached the age of 110 in 2010, reflecting a >100-fold difference in survival probability[11]. Although the number of cases is still very limited, observational studies have reported distinctive characteristics of supercentenarians, that is, an exceptionally long and healthy lifespan with physical independence and relatively high cognitive function beyond the age of 100 years[12–14]. Furthermore, a demographic debate over the existence of a fixed maximal human lifespan has recently stimulated scientific inquiries on supercentenarians[15,16]. However, the biological mechanisms that permit the oldest individuals to reach the ceiling of human longevity remain poorly characterized.

We hypothesize that survival into the current highest age at death (≥110 years) in low-mortality countries is supported by protection against cardiovascular disease, as this system is intrinsically more susceptible to oxidative stress and inflammation with aging, and plays a central role in maintaining oxygen and metabolite delivery to major organ systems. To test this hypothesis, we select nine circulating biomarkers reflecting distinct cardioprotective and pathogenic pathways on the basis of previous epidemiological evidence and biological rationale[17–20]. Four biomarkers of endogenous cardioprotective molecules include N-terminal pro-B-type natriuretic peptide (NT-proBNP, neurohormonal activity), erythropoietin (erythropoiesis and hypoxic response mediated by hypoxia-inducible factor-1 (HIF1)), adiponectin (insulin-sensitizing and anti-inflammatory pathway), and extracellular superoxide dismutase (EC-SOD or SOD3, antioxidant enzyme in the arterial wall). B-type natriuretic peptide, a bioactive counterpart of NT-proBNP, causes natriuresis and diuresis, arterial dilatation, and antagonism of the renin–angiotensin–aldosterone system, thus counter-regulating hemodynamic abnormalities in heart failure[21]. All of these cardioprotective biomarkers are upregulated in elderly patients with heart failure[17–20]. Three inflammatory mediators include interleukin-6, tumor necrosis factor-alpha (TNF-alpha), and angiopoietin-like protein 2 (Angptl2). Angptl2 is upregulated in obesity and type 2 diabetes and accelerates endothelial inflammation, atherosclerosis, and the pathogenesis of heart failure[22,23]. Finally, reduced reserve capacity of multiple organ systems is involved in heart failure in old age[24]; hence, the levels of two biomarkers, cystatin C and cholinesterase, are measured as indicators of the functional reserves of the kidney and liver, respectively[25,26]. Cystatin C is selected because it shows a much higher correlation with age than does creatinine in approximately 5000 healthy individuals ranging from 25 to 110 years[27]. These nine candidate biomarkers are assessed for associations with survival in multiple cohorts of centenarians, (semi)-supercentenarians, and very old individuals, compared with traditional cardiovascular risk factors and plasma albumin, which are independent predictors of mortality in older adults[28]. First, we show an age-related increase in cardioprotective and inflammatory biomarkers, and a decrease in organ reserves up to 115 years of age. Of these, four biomarkers including NT-proBNP, interleukin-6, cystatin C, and cholinesterase are associated with all-cause mortality at the oldest old. Finally, only the relationship between NT-proBNP and all-cause mortality is robust against adjustment for traditional risk factors, inflammation, and organ reserve.

## Results

**Study population.** For this study, we aggregated data from three prospective cohort studies: the Tokyo Centenarian Study (TCS)[29,30], Japanese Semi-supercentenarian Study (JSS)[14,31], and Tokyo Oldest Old Survey on Total Health (TOOTH)[32]. The analytic cohort comprised 1427 oldest-old individuals, including 36 supercentenarians (aged ≥110 years at enrollment), 572 semi-supercentenarians (105–109 years), 288 centenarians (100–104 years), and 531 very old persons (85–99 years). The selected characteristics of the participants are shown in Table 1 (the full version is shown in Supplementary Table 1). Centenarians show distinctive cardiometabolic profiles: a low prevalence of diabetes mellitus, as opposed to a relatively high prevalence of moderate to severe chronic kidney disease (i.e. stage 3b-5). The majority of supercentenarians (56.3%) did not take any cardiovascular medication (Supplementary Table 1). The most common ECG findings in supercentenarians were first-degree atrioventricular block (31.0%) and non-specific ST-T change (27.6%), followed by left anterior hemiblock (20.7%, Supplementary Table 1).

A common missense variant in codon 213 in exon 3 of SOD3 (rs1799895) has been shown to be associated with an approximately 10-fold increase in plasma EC-SOD concentrations and an elevated risk for incidental ischemic heart disease[33]. SOD3 R213G (rs1799895) variation did not significantly differ across age groups ($P = 0.661$, Pearson's Chi-Squared test). Plasma EC-SOD concentrations were 9.4-fold and 24-fold higher in rs1799895 heterozygotes and homozygotes, respectively, compared to those in non-carriers (Fig. 1). Thus, only individuals without SOD3 R213G variants (non-carriers) were included in subsequent analyses.

**Relationship between circulating biomarkers and age by cardiovascular status.** Cross-sectionally, plasma levels of endogenous cardioprotective molecules and inflammatory mediators, except Angptl2, were correlated with age up to 115 years (Fig. 2). Of these, only NT-proBNP showed age-related distributions, with or without a cardiovascular abnormality (Fig. 2a), as well as an association with the degree of ECG abnormality (Supplementary Fig. 1). Correlations between circulating biomarkers and age were largely maintained, regardless of cardiovascular status, when the

**Table 1 Selected characteristics of participants according to age at enrollment.**

| Characteristics | Very old | | Centenarians | | Semi-supercentenarians | | Supercentenarians | | P for trend |
|---|---|---|---|---|---|---|---|---|---|
| | **N** | **(85–99 years)** | **N** | **(100–104 years)** | **N** | **(105–109 years)** | **N** | **(110 + years)** | |
| Female, no. (%) | 531 | 298 (56.1%) | 288 | 225 (78.1%) | 572 | 502 (87.8%) | 36 | 34 (94.4%) | <0.001 |
| Current smoker, no. (%) | 511 | 36 (7.1%) | 282 | 3 (1.1%) | 564 | 7 (1.2%) | 35 | 1 (2.9%) | <0.001 |
| High education, no. (%) | 513 | 193 (37.6%) | 275 | 61 (22.2%) | 540 | 63 (11.7%) | 34 | 3 (8.8%) | <0.001 |
| Body mass index, kg/m$^2$ | 528 | 21.5 ± 3.2 | 187 | 19.5 ± 3.2 | 353 | 19.4 ± 3.3 | 21 | 18.4 ± 2.9 | <0.001 |
| *Medical history* | | | | | | | | | |
| CHD, no. (%) | 531 | 53 (10.0%) | 283 | 41 (14.5%) | 566 | 78 (13.8%) | 36 | 3 (8.3%) | 0.124 |
| Stroke, no. (%) | 531 | 92 (17.3%) | 283 | 46 (16.3%) | 566 | 123 (21.7%) | 36 | 2 (5.6%) | 0.268 |
| Hypertension, no. (%) | 531 | 334 (62.9%) | 287 | 110 (38.3%) | 568 | 254 (44.7%) | 36 | 14 (38.9%) | <0.001 |
| Hyperlipidemia, no. (%) | 530 | 251 (47.4%) | 288 | 40 (13.9%) | 572 | 83 (14.5%) | 36 | 8 (22.2%) | <0.001 |
| Diabetes mellitus, no. (%) | 531 | 99 (18.6%) | 288 | 21 (7.3%) | 572 | 32 (5.6%) | 36 | 2 (5.6%) | <0.001 |
| CKD (stage 3b-5), no. (%) | 530 | 77 (14.5%) | 288 | 101 (35.1%) | 572 | 214 (37.4%) | 36 | 11 (30.6%) | <0.001 |
| *Medication* | | | | | | | | | |
| Nitrate, no. (%) | 527 | 53 (10.1%) | 279 | 39 (14.0%) | 561 | 79 (14.1%) | 32 | 3 (9.4%) | 0.084 |
| Oral anticoagulant, no. (%) | 527 | 20 (3.8%) | 279 | 1 (0.4%) | 561 | 6 (1.1%) | 32 | 0 (0.0%) | <0.001 |
| Antiarrhythmics, no. (%) | 527 | 21 (4.0%) | 279 | 3 (1.1%) | 561 | 9 (1.6%) | 32 | 0 (0.0%) | 0.007 |
| Digoxin, no. (%) | 527 | 16 (3.0%) | 279 | 11 (3.9%) | 561 | 32 (5.7%) | 32 | 1 (3.1%) | 0.050 |
| Diuretics, no. (%) | 527 | 61 (11.6%) | 279 | 62 (22.2%) | 561 | 166 (29.6%) | 32 | 9 (28.1%) | <0.001 |
| CCB, no. (%) | 527 | 213 (40.4%) | 279 | 47 (16.9%) | 561 | 101 (18.0%) | 32 | 3 (9.4%) | <0.001 |
| ACEI or ARB, no. (%) | 527 | 157 (29.8%) | 279 | 26 (9.3%) | 561 | 70 (12.5%) | 32 | 6 (18.8%) | <0.001 |
| Statin, no. (%) | 527 | 81 (15.4%) | 279 | 5 (1.8%) | 561 | 10 (1.8%) | 32 | 1 (3.1%) | <0.001 |
| *Electrocardiogram* | | | | | | | | | |
| Normal, no. (%) | 521 | 151 (29.0%) | 193 | 41 (21.2%) | 453 | 57 (12.6%) | 29 | 4 (13.8%) | <0.001 |
| OMI, no. (%) | 521 | 21 (4.0%) | 193 | 8 (4.2%) | 453 | 52 (11.5%) | 29 | 4 (13.8%) | <0.001 |
| Pacemaker, no. (%) | 521 | 6 (1.2%) | 193 | 3 (1.6%) | 453 | 5 (1.1%) | 29 | 2 (6.9%) | 0.409 |
| Atrial fibrillation, no. (%) | 521 | 23 (4.4%) | 193 | 13 (6.7%) | 453 | 29 (6.4%) | 29 | 1 (3.5%) | 0.257 |
| LVH, no. (%) | 521 | 90 (17.3%) | 193 | 21 (10.9%) | 453 | 56 (12.4%) | 29 | 1 (3.5%) | 0.008 |
| AAB, no. (%) | 521 | 0 (0.0%) | 193 | 2 (1.0%) | 453 | 5 (1.1%) | 29 | 0 (0.0%) | 0.045 |
| *Cardioprotective factors* | | | | | | | | | |
| NT-proBNP, pg/mL (IQR) | 475 | 195 (115–392) | 199 | 687 (376–1360) | 385 | 960 (465–1900) | 21 | 1530 (587–2540) | <0.001 |
| EPO, mIU/mL (IQR) | 415 | 10.3 (7.8–14.2) | 199 | 10.4 (7.6–14.4) | 385 | 11.3 (8.2–16.3) | 21 | 12.2 (8.4–15.2) | 0.031 |
| *SOD3 R213G genotype (rs1799895)* | | | | | | | | | |
| Non-carrier, no. (%) | 530 | 475 (89.6%) | 288 | 265 (92.0%) | 565 | 518 (91.7%) | 35 | 33 (94.3%) | 0.661[b] |
| Heterozygotes, no. (%) | 530 | 51 (9.6%) | 288 | 20 (6.9%) | 565 | 45 (8.0%) | 35 | 2 (5.7%) | |
| Homozygotes, no. (%) | 530 | 4 (0.8%) | 288 | 3 (1.0%) | 565 | 2 (0.4%) | 35 | 0 (0.0%) | |
| EC-SOD [a], ng/mL (IQR) | 448 | 106 (88–127) | 222 | 137 (113–169) | 324 | 146 (121–180) | 20 | 168 (124–203) | <0.001 |
| ADPN, ng/mL (IQR) | 529 | 12.0 (7.3–19.2) | 271 | 16.9 (12.5–23.2) | 537 | 18.5 (13.3–25.0) | 34 | 20.2 (16.4–23.9) | <0.001 |
| *Inflammatory mediators* | | | | | | | | | |
| Interleukin-6, pg/mL (IQR) | 529 | 1.7 (1.3–2.5) | 272 | 2.9 (2.3–4.3) | 545 | 3.4 (2.3–5.4) | 34 | 4.9 (3.0–7.3) | <0.001 |
| TNF-alpha, pg/mL (IQR) | 529 | 2.2 (1.9–2.8) | 272 | 3.4 (2.7–4.2) | 536 | 4.2 (3.0–5.6) | 32 | 3.9 (2.5–4.9) | <0.001 |
| Angptl2, ng/mL (IQR) | 529 | 4.1 (3.2–5.3) | 252 | 3.9 (3.2–5.0) | 409 | 4.2 (3.4–5.1) | 25 | 4.1 (3.5–5.1) | 0.461 |
| *Organ reserve* | | | | | | | | | |
| Cystatin C, mg/dL | 524 | 1.26 ± 0.51 | 265 | 1.63 ± 0.52 | 522 | 1.80 ± 0.54 | 31 | 1.84 ± 0.60 | <0.001 |
| Cholinesterase, IU/L | 531 | 277 ± 68 | 287 | 214 ± 56 | 569 | 196 ± 58 | 36 | 177 ± 42 | <0.001 |
| *Traditional risk factors (continuous variables)* | | | | | | | | | |
| HDL-C, mg/dL | 530 | 58.8 ± 14.7 | 288 | 52.5 ± 13.3 | 572 | 45.8 ± 11.9 | 36 | 45.9 ± 11.8 | <0.001 |
| LDL-C, mg/dL | 525 | 117.7 ± 26.9 | 288 | 99.8 ± 28.2 | 572 | 102.0 ± 28.1 | 36 | 96.2 ± 29.3 | <0.001 |
| Creatinine, mg/dL | 530 | 0.84 ± 0.51 | 288 | 0.90 ± 0.45 | 572 | 0.87 ± 0.43 | 36 | 0.82 ± 0.34 | 0.684 |
| CRP, mg/dL (IQR) | 531 | 0.09 (0.04–0.19) | 287 | 0.16 (0.05–0.46) | 572 | 0.25 (0.09–0.66) | 36 | 0.34 (0.13–0.88) | <0.001 |
| Albumin, g/dL | 531 | 4.1 ± 0.3 | 287 | 3.6 ± 0.4 | 572 | 3.4 ± 0.4 | 36 | 3.2 ± 0.4 | <0.001 |

Plus-minus values are means ± SD. Trends in each characteristic of participants across four age groups were analyzed using the trend test for continuous variables, and the Cochran-Armitage test for trend for categorical variables.

*IQR* interquartile range, *CHD* coronary heart disease, *CKD* chronic kidney disease, *CCB* calcium-channel blocker, *ACEI* angiotensin-converting enzyme inhibitor, *ARB* Angiotensin II Receptor Blocker, *OMI* old myocardial infarction, *LVH* left ventricular hypertrophy, *AAB* advanced atrioventricular block, *NT-proBNP* N-terminal pro-brain natriuretic peptide, *EPO* erythropoietin, *EC-SOD* extracellular superoxide dismutase, *ADPN* adiponectin, *TNF-alpha* tumor necrosis factor-alpha, *Angptl2* angiopoietin-like protein 2, *HDL-C* high-density lipoprotein-cholesterol, *LDL-C* low-density lipoprotein-cholesterol, *CRP* C-reactive protein.
[a]Only individuals with RR genotype in *SOD3* (rs1799895) were included in analysis.
[b]P value for Pearson's Chi-squared test.

analytic cohort was restricted to only female participants (Supplementary Table 2).

Plasma NT-proBNP levels, adjusted for age and sex, were modestly correlated with each biomarker, except Angptl2 (Supplementary Table 3; the highest correlation was with cystatin C ($r = 0.394$, $P < 0.001$). The findings were largely similar in participants without a cardiovascular abnormality at baseline (Supplementary Table 3). Multivariate stepwise linear regression with backward elimination showed that age, cystatin C, and major ECG abnormality, namely, atrial fibrillation, were substantially associated with NT-proBNP ($R^2 = 0.580$, adjusted $R^2 = 0.572$ for the model, Supplementary Table 4).

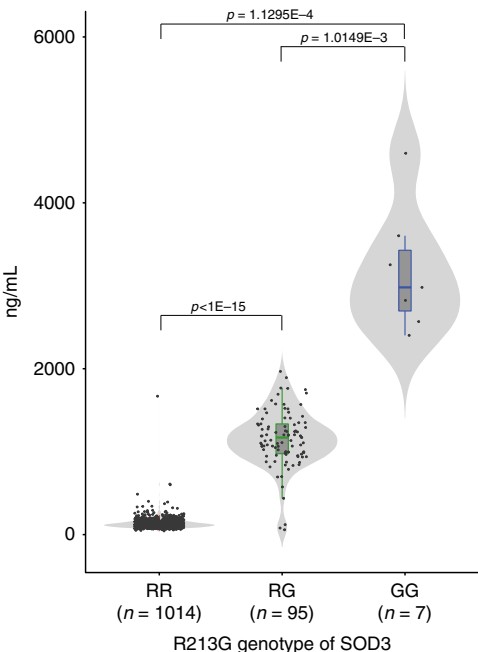

**Fig. 1 Plasma extracellular superoxide dismutase concentrations by *SOD3* R213G variant (rs1799895).** Plasma concentrations of extracellular superoxide dismutase (EC-SOD) were measured in R213G non-carriers (RR, $N = 1014$), R213G heterozygotes (RG, $N = 95$), and R213G homozygotes (GG, $N = 7$). As compared to non-carriers, EC-SOD concentrations were significantly higher in heterozygotes or homozygotes ($P < 1E{-}15$, $P = 1.1295E{-}4$, respectively), Violin plots show plasma concentrations of EC-SOD by *SOD3* R213G variants. Boxes denote interquartile range, the horizontal lines denote medians, and whiskers denote 10th (lower) and 90th (upper) percentiles. *P* values are from one-way ANOVA with post hoc Bonferroni test.

**Associations between biomarkers and mortality**. During follow-up, 1000 of 1427 participants (70.1%) died, with 191 deaths in the very old cohort (7.8 per 100 person-years), 270 deaths among centenarians aged 100–104 years at baseline (36.2 per 100 person-years), 506 deaths among semi-supercentenarians (58.2 per 100 person-years), and 33 deaths among supercentenarians (74.3 per 100 person-years).

Figure 3 is a flowchart showing the statistical analysis procedures used to identify specific biomarkers associated with exceptional survival. First, we constructed a base model with risk factors for old age mortality (Supplementary Table 5). We thus calculated overall and age-group specific hazard ratios (HR) for all-cause mortality according to each candidate biomarker (as continuous variables) using univariable and multivariable Cox proportional models (Fig. 4). In both overall and age-stratified cohorts, high levels of NT-proBNP, interleukin-6, and cystatin C, and low levels of cholinesterase were associated with an increased risk of all-cause mortality, independent of traditional risk factors (Fig. 4). Among these, only NT-proBNP was associated consistently with mortality across age groups. Adiponectin was associated significantly with all-cause mortality in the very old and in centenarians aged 100–104 years, however, the direction of associations was opposite between the two cohorts. Thus, we excluded this biomarker from further survival analysis.

Of the traditional risk factors treated as continuous variables, creatinine, C-reactive protein (CRP), and albumin were associated with all-cause mortality in both the total and age-stratified cohorts, whereas high- and low-density lipoprotein cholesterol, hemoglobin A1c, and estimated glomerular filtration rate by

serum creatinine (eGFR-cr) were not (Fig. 5). Only low albumin was associated with high mortality across age groups.

To identify the best overall set of predictors of mortality, the prognostic biomarkers identified in Fig. 4 (NT-proBNP, interleukin-6, cystatin C, and cholinesterase) and Fig. 5 (creatinine, CRP, and albumin) were combined with clinical covariates in the base model. The least absolute shrinkage and selection operator (LASSO) with five-fold cross-validation was used for variable selection (Fig. 6a–d, and Supplementary Table 6), followed by multivariate Cox regression to identify the final set of prognostic markers (Fig. 6e). In addition to clinical predictors, such as age, sex, and a history of cardiovascular disease and hypertension (negative risk), we determined that NT-proBNP, cholinesterase, and albumin were among the most significant prognostic markers in the entire cohort. NT-proBNP was a significant prognostic marker in centenarians (≥100 years old), and associations between low NT-proBNP and survival advantage were robust in individuals aged ≥105 years in both the forward stepwise regression and forced entry models (Supplementary Table 7 and Supplementary Table 8). Albumin was the only biomarker that was among the most significant prognostic markers across age group and regardless of statistical models (Fig. 6, Supplementary Table 7, and Supplementary Table 8). In the highest age group (≥105 years), major ECG abnormality was among the most significant prognostic markers in the LASSO-Cox model (Fig. 6e) as well as the forward stepwise regression model (Supplementary Table 7), which was mostly attributable to atrial fibrillation and old myocardial infarction (HR, 1.75, 95% CI, 1.03–2.98, $P = 0.038$ and HR, 1.63, 95% CI, 1.09–2.43, $P = 0.017$ for atrial fibrillation and old myocardial infarction, respectively, Supplementary Table 8).

In categorical analysis with a conventional cutoff point of heart failure, an NT-proBNP level ≥1,800 pg/mL was significantly associated with all-cause mortality in individuals aged ≥105 years, but not in individuals aged 85–99 years, or 100–104 years, because of the relatively small number of such individuals with high levels of NT-proBNP in younger cohorts (Supplementary Table 9).

In sensitivity analysis restricted to those without a cardiovascular abnormality, cystatin-C was significantly associated with all-cause mortality in individuals aged ≥105 years, while NT-proBNP and interleukin-6 remained associations across age groups (Supplementary Fig. 2). The second sensitivity analysis, where individuals with age-specific highest tertile of cystatin-C were excluded from the analysis, associations between NT-proBNP and all-cause mortality were attenuated but remained significant in the highest age group (≥105 years, Supplementary Fig. 3).

Data on cause-specific mortality were available for the very old, aged 85–99 years; 48 died from cardiovascular disease, 41 from cancers, 36 from pneumonia, 5 from advanced dementia, 34 from other causes, and 27 from unknown causes. NT-proBNP and interleukin-6 were significantly associated only with cardiovascular mortality (HR, 1.52; 95% CI, 1.08–2.16 and HR, 1.33; 95% CI, 1.02–1.74 in the multivariate model, respectively in Fig. 7), while cholinesterase and albumin were significantly associated only with non-cardiovascular mortality (HR, 0.57; 95% CI, 0.44–0.73 and HR, 0.70, 95% CI, 0.57–0.87 in the multivariate model, respectively). Cystatin C was significantly associated with both cardiovascular and non-cardiovascular mortality (Fig. 7).

**Prognostic performances of biomarkers**. In order to assess the prognostic performance of each biomarker, we computed Concordance index (C-index) based on Cox proportional hazard models[34]. In the entire cohort, the addition of each prognostic biomarker in Fig. 4 (NT-proBNP, interleukin-6, cystatin C, and

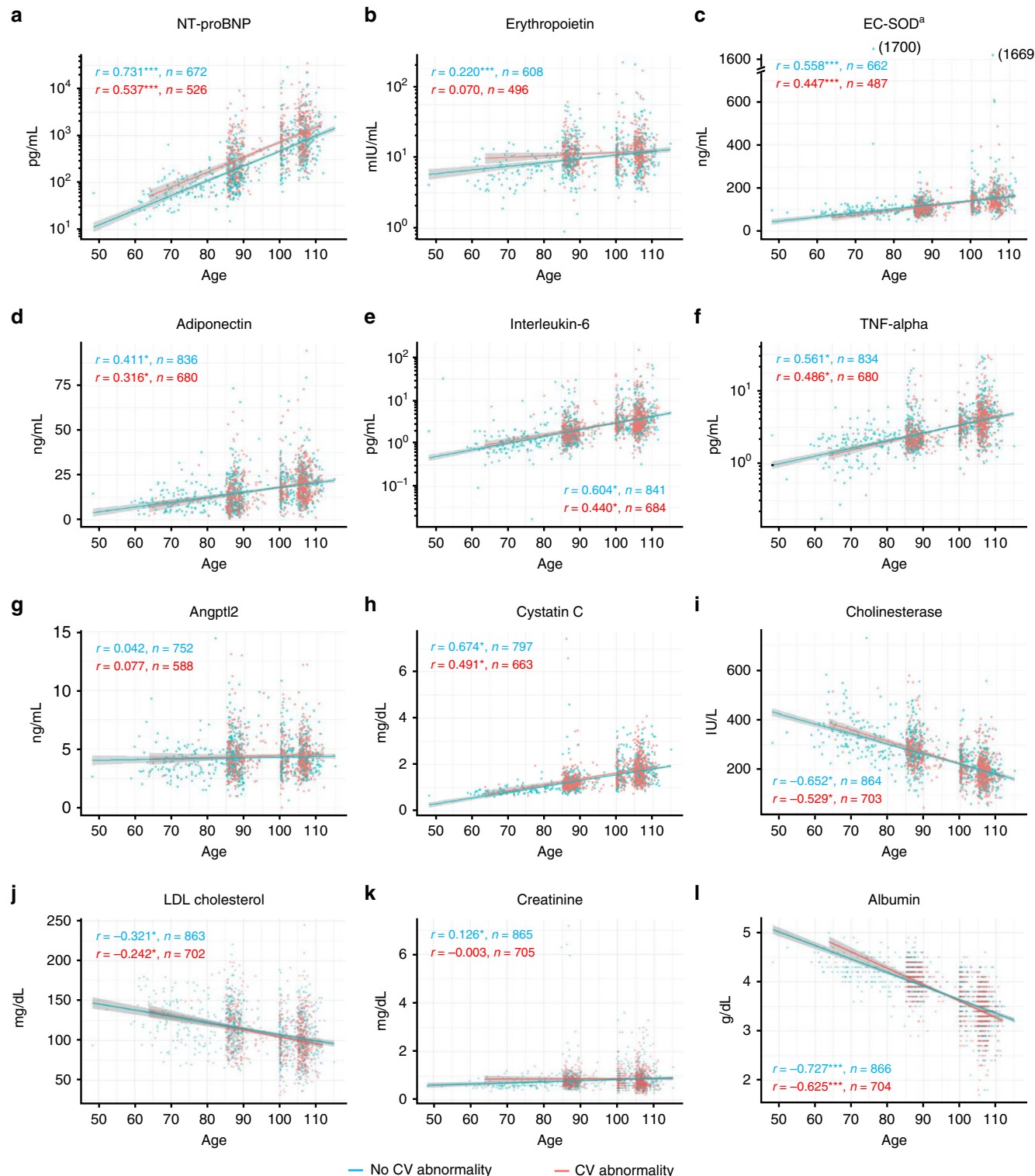

— No CV abnormality    — CV abnormality

cholinesterase) to the base model significantly improved risk prediction with relatively small optimisms (Table 2). When stratified by age, the predictivity of the models substantially declined, suggesting that age itself is a dominant prognostic factor. The base model is only weakly predictive of mortality beyond 105 years of age (C-index, 0.617 [95% CI; 0.577–0.656], optimism-corrected C-index = 0.588). Nevertheless, adding NT-proBNP and, to a lesser degree, cholinesterase significantly improve the predictivity at the highest ages (NT-proBNP: C-index, 0.653 [95% CI; 0.615–0.691], $P = 0.001$, optimism-corrected C-index = 0.625; Cholinesterase:

C-index, 0.636 [95% CI; 0.596–0.676], $P = 0.019$, optimism-corrected C-index = 0.609, respectively, Table 2).

**Biological correlates of exceptional survival to supercentenarian age.** During follow-up, 123 participants (11 males, 112 female) died at age ≥110 years (decedent supercentenarians), 183 participants died at age 100–104 years (decedent centenarians), and 504 participants died at age 105–109 years (decedent semi-supercentenarians). The relationships between the identified prognostic biomarkers and age at enrollment across these three

**Fig. 2 Cross-sectional associations between circulating biomarkers and age by cardiovascular status.** Scatter plots show cross-sectional associations between biomarkers of cardioprotective pathways (**a–d**), inflammation (**e–g**), organ reserve (**h**, **i**), and selected traditional risk factors (**j–l**) and age at enrollment, according to presence (red) or absence (blue) of a cardiovascular abnormality. All the biomarkers were assessed at the time of enrollment. Spearman's correlation coefficients between biomarkers and age at enrollment and sample numbers are shown for those with (red) or without (blue) cardiovascular abnormality. The solid lines represent the correlation lines, and the shaded area represents the 95% confidence interval of the correlation line. Unrelated family members of the centenarians (spouses of the first-degree offspring of the centenarians) aged between 48 and 94 years (mean age, 73.1 years) were included as a younger control group (n = 167 at the maximum). Characteristics of this population are described in ref. [31]. Population sizes for the twelve biomarkers differ due to variation in the bio-banking of samples. Participants were considered to have a cardiovascular abnormality when one or more of the following criteria were fulfilled: (1) a history of coronary heart disease or stroke, (2) cardiovascular medication use (i.e., nitrate, oral anticoagulant, antiarrhythmic drug, or digoxin), and (3) a major electrocardiographic abnormality (Table 1 and Supplementary Table 1). Classification of cardiovascular abnormality in unrelated family of centenarians was based on medical history and medication list because of lack of ECG assessment in this population. *NT-proBNP* indicates N-terminal pro-brain natriuretic peptide, *EC-SOD* extracellular superoxide dismutase, *TNF*-alpha tumor necrosis factor-alpha, *Angptl2* angiopoietin-like protein 2. ᵃ Only individuals with 213RR genotype (non-carrier) in *SOD3* (rs1799895) were included in the analysis. *P < 0.001.

decedent centenarian categories were evaluated (Fig. 8). Only NT-proBNP showed age-specific distributions capable of distinguishing decedent supercentenarians from younger cohorts (Fig. 8a). These findings were statistically confirmed by a cross-tabular analysis: NT-proBNP levels were significantly lower in decedent supercentenarians than in other decedent centenarians at any given age at assessment (Supplementary Table 10).

## Discussion

Theoretically, supercentenarians are able to approach the current human longevity limit by preventing or surviving major illness, including cardiovascular disease. Identifying specific biomarkers conducive to exceptional survival might provide molecular insights into the protective mechanisms against aging-related disease. Our results highlight molecular pathways common to both extended longevity and manifestations of heart failure, which are characterized by concomitant increases in endogenous cardioprotective factors and inflammatory mediators, accompanied by reductions in major organ reserves. Despite these indicators, the longest-lived individuals exhibited a relative absence of overt cardiovascular disease and 56.3% of the supercentenarians in the present study were not taking cardiovascular medications. We identified four prognostic biomarkers associated with all-cause mortality among the oldest old, independent of traditional cardiovascular risk factors and plasma albumin: NT-proBNP, interleukin-6, cystatin-C, and cholinesterase. Among these, only the association between low NT-proBNP levels and survival advantage was robust against adjustment for other prognostic biomarkers, especially in those aged >105 years, and consistent results in the sensitivity and retrospective analyses were observed. Collectively, the present study is the first to demonstrate that a low concentration of NT-proBNP in the circulation is more relevant for exceptional survival to the highest ages than traditional cardiovascular risk factors, inflammation, and organ reserve.

NT-proBNP is a cleavage product of the prohormone proBNP, which is synthesized predominantly in the left ventricle myocyte in reaction to ventricular wall stress and myocardial ischemia[35,36]. Plasma NT-proBNP levels provide prognostic information in a variety of cardiovascular settings, including heart failure with reduced[37] or preserved ejection fraction[38], coronary heart disease[39], and atrial fibrillation[40], indicating the clinical utility of this molecule as a biomarker for myocyte stress[21]. Moreover, circulating NT-proBNP predicts all-cause and cardiovascular mortality in the elderly and nonagenarians without prevalent cardiovascular disease[41,42], suggesting that this molecule also reflects structural and functional cardiac disorders at the subclinical stage. The plasma NT-proBNP level also increases with both renal dysfunction and aging in patients with or without

heart failure[43]. Our findings that the circulating NT-proBNP levels strongly correlated with age, cystatin C, and atrial fibrillation corroborate well with previous epidemiological observations suggesting that determinants of circulating NT-proBNP in the oldest old are similar to those in the younger population.

Underlying mechanisms responsible for association between low levels of circulating NT-proBNP and exceptional survival are not clear at this time, but potential explanation can be considered. In the present study, centenarians and (semi-)supercentenarians showed a low prevalence of clinical and subclinical cardiovascular disease detectable by ECG, as well as low cardiometabolic risk profiles, except for a relatively high prevalence of chronic kidney disease. Nevertheless, the median NT-proBNP level continuously increased with age regardless of cardiovascular status, reaching 1530 pg/mL in those aged ≥110 years. Furthermore, 42.9% of supercentenarians had a NT-proBNP level ≥1800 pg/mL, which is the cutoff point for the diagnosis of heart failure in individuals aged >75 years[44]. In our study, both NT-proBNP and cystatin C were significantly associated with cardiovascular mortality in the very old, and with all-cause mortality in the total and the highest age group. When individuals with highest tertile of cystatin C were excluded, associations between NT-proBNP and mortality were attenuated in younger cohorts but remained significant in those aged ≥105 years. These results suggest that upregulation of NT-proBNP in asymptomatic centenarians reflects a compensatory homeostatic response to hemodynamic stress, which arises from the interplay between cardiovascular and, potentially, renal alterations associated with advanced age, and ultimately limiting chances of survival to the supercentenarian age. Despite its low prevalence (6.4% in those aged 105–109 years and 3.5% in those aged 110 years or older), atrial fibrillation on ECG has a significant prognostic impact in the highest age group, suggesting potential hemodynamic vulnerability in this cohort. Our interpretation is supported by previous reports that diuretics are the most commonly used drug among Swedish[45] and Danish[46] centenarians, although these groups acknowledged that cardiovascular medications were relatively under prescribed. Future studies incorporating detailed assessment of cardiac structure and function using ultrasound cardiography are warranted to further test our interpretation.

Identifying cause of death of centenarians is challenging, because a significant part of this population dies in the non-hospital setting (e.g., nursing home or residential care home) and with asymptomatic clinical presentations. In a population-based study using the death registration of 35,867 centenarians who died in England between 2001 and 2010 (GUIDE Care project), the most prevalent cause of death was old age (28.1%), followed by pneumonia (17.7%), cerebrovascular disease (10.0%), and other circulatory diseases (9.8%). Death from ischemic heart

<Main analysis>

STEP 0. Building a base model by using risk factors for old age mortality

・Demographic and lifestyle factors: age (continuous), sex, education, current smoking
・Traditional cardiovascular risk factors: history of cardiovascular disease, hypertension, hyperlipidemia, diabetes, mellitus, chronic kidney disease (stage 3b-5), elevated CRP (≥0.3mg/dL), major ECG abnormality, cardiovascular medication
・Low plasma albumin (<3.5g/dL)

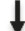

STEP 1. Identification of prognostic factors from 9 candidate biomarkers (continuous variables) using Cox proportional hazard model (the base model)

・Cardioprotective factors: NT-proBNP, erythropoietin, adiponectin, EC-SOD
・Inflammatory mediators: Interleukin-6, TNF-alpha, Angptl2
・Markers for organ reserve: Cystatin C, Cholinesterase

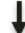

STEP 2. Identification of prognostic factors from traditional risk factors (continuous variables) using Cox proportional hazard model (the base model)

・HDL-and LDL-cholesterol, Hemoglobin A1c, Creatinine, eGFRcr, CRP and albumin

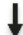

STEP 3. Selection of the best overall predictor of mortality from prognostic biomarkers Identified in STEP 1 and STEP 2 and clinical covariates in the base models by using (1) LASSO Cox, (2) Forward stepwise regression, (3) Forced entry model

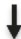

STEP 4. C-index for discrimination of all-cause mortality with optimism correction

STEP 5. Retrospective analysis for associations between the prognostic biomarkers and age at assessment according to deceased centenarian categories

<Sub analysis>

S1. Categorical analysis of the candidate biomarkers with all-cause mortality (the base model)

S2. Sensitivity analysis for the associations between candidate biomarkers and mortality, with exclusion of (1) participants with cardiovascular abnormality, or (2) those with age-specific highest tertile of cystatin C

S3. Cardiovascular and non-cardiovascular mortality with candidate biomarkers and traditional risk factors (continuous variables) in the vey old cohort (85–99 years only)

**Fig. 3 Flowchart of statistical analysis to identify specific biomarkers associated with exceptional survival.** *NT-proBNP* N-terminal pro-brain natriuretic peptide, *EC-SOD* extracellular superoxide dismutase, *TNF-alpha* tumor necrosis factor-alpha, *Angptl2* angiopoietin-like protein 2.

disease (8.6%) and cancers (4.4%) are uncommon compared to that among people of younger old age (80–84 years)[47]. Autopsy is the best possible method to accurately identify the cause of death in the oldest old. In an autopsy study of 40 centenarians who died unexpectedly out of hospital, the most common cause of death was cardiovascular disease (68%), respiratory disease (25%), gastrointestinal disease (5%), and cerebrovascular disease (2%), but no centenarians died from cancer[48]. Another autopsy study of 140 centenarians and 96 older adults aged 75–95 years found that the prevalence of ischemic cardiomyopathies was equivocal (37.8% and 33.3%, in centenarians and older adults, respectively)

but that of acute myocardial infarction was lower (5.9% and 20.5%, $P = 0.001$) and that of cardiac amyloidosis was higher (11.3% and 0.0%, $P = 0.002$) in centenarians than in older adults[49]. Given the low cardiometabolic risk in this population, the high NT-proBNP levels of centenarians may reflect cardiac dysfunction attributable to age-related myocardial remodeling. Supercentenarians, by virtue of a postponed age-related increase in circulating NT-proBNP, may be equipped with efficient physiological mechanisms for delaying the processes of cardiovascular aging, which involve mitochondrial oxidative stress, dysfunctional autophagy and ubiquitin proteasome system,

| Biomarker | Crude | | Adjusted | | |
|---|---|---|---|---|---|
| | HR (95% CI) | p | HR (95% CI) | p | |
| **Entire cohort** | | | | | |
| NT-proBNP | 2.04 (1.89–2.19) | <0.001 | 1.34 (1.19–1.51) | <0.001 | |
| Erythropoietin | 1.10 (1.01–1.19) | 0.025 | 1.04 (0.94–1.14) | 0.466 | |
| EC-SOD[a] | 1.46 (1.36–1.56) | <0.001 | 1.03 (0.93–1.15) | 0.550 | |
| Adiponectin | 1.37 (1.29–1.47) | <0.001 | 1.06 (0.98–1.16) | 0.152 | |
| Interleukin-6 | 1.62 (1.54–1.71) | <0.001 | 1.18 (1.09–1.27) | <0.001 | |
| TNF-alpha | 1.49 (1.41–1.57) | <0.001 | 1.08 (0.99–1.17) | 0.081 | |
| Angptl2 | 1.04 (0.97–1.11) | 0.311 | 0.99 (0.91–1.09) | 0.914 | |
| Cystatin C | 1.40 (1.35–1.46) | <0.001 | 1.31 (1.18–1.45) | <0.001 | |
| Cholinesterase | 0.43 (0.40–0.47) | <0.001 | 0.74 (0.66–0.83) | <0.001 | |
| | | | | | |
| **85–99 years** | | | | | |
| NT-proBNP | 1.29 (1.11–1.50) | 0.001 | 1.26 (1.05–1.51) | 0.013 | |
| Erythropoietin | 1.01 (0.86–1.20) | 0.883 | 1.02 (0.85–1.23) | 0.804 | |
| EC-SOD[a] | 0.99 (0.84–1.15) | 0.855 | 1.05 (0.87–1.26) | 0.630 | |
| Adiponectin | 1.16 (1.00–1.34) | 0.045 | 1.20 (1.01–1.42) | 0.035 | |
| Interleukin-6 | 1.29 (1.13–1.47) | <0.001 | 1.22 (1.05–1.41) | 0.011 | |
| TNF-alpha | 1.11 (0.98–1.26) | 0.093 | 1.11 (0.96–1.27) | 0.158 | |
| Angptl2 | 1.12 (0.96–1.29) | 0.139 | 0.99 (0.85–1.16) | 0.910 | |
| Cystatin C | 1.22 (1.11–1.35) | <0.001 | 1.33 (1.15–1.53) | <0.001 | |
| Cholinesterase | 0.65 (0.54–0.77) | <0.001 | 0.68 (0.56–0.83) | <0.001 | |
| | | | | | |
| **100–104 years** | | | | | |
| NT-proBNP | 1.46 (1.25–1.72) | <0.001 | 1.29 (1.01–1.65) | 0.042 | |
| Erythropoietin | 1.12 (0.94–1.33) | 0.210 | 1.07 (0.87–1.32) | 0.503 | |
| EC-SOD[a] | 1.05 (0.91–1.20) | 0.510 | 0.99 (0.81–1.22) | 0.957 | |
| Adiponectin | 0.96 (0.85–1.08) | 0.488 | 0.82 (0.70–0.96) | 0.015 | |
| Interleukin-6 | 1.20 (1.06–1.36) | 0.004 | 1.09 (0.92–1.29) | 0.321 | |
| TNF-alpha | 1.15 (1.03–1.28) | 0.012 | 1.11 (0.90–1.35) | 0.328 | |
| Angptl2 | 0.96 (0.84–1.11) | 0.608 | 0.92 (0.77–1.10) | 0.359 | |
| Cystatin C | 1.20 (1.06–1.35) | 0.004 | 1.26 (0.96–1.65) | 0.101 | |
| Cholinesterase | 0.70 (0.61–0.81) | <0.001 | 0.90 (0.72–1.11) | 0.324 | |
| | | | | | |
| **105 years or older** | | | | | |
| NT-proBNP | 1.33 (1.20–1.47) | <0.001 | 1.34 (1.17–1.53) | <0.001 | |
| Erythropoietin | 1.09 (0.97–1.23) | 0.132 | 1.05 (0.92–1.20) | 0.430 | |
| EC-SOD[a] | 0.98 (0.88–1.10) | 0.737 | 1.02 (0.88–1.19) | 0.812 | |
| Adiponectin | 1.07 (0.98–1.17) | 0.126 | 1.07 (0.96–1.20) | 0.216 | |
| Interleukin-6 | 1.26 (1.16–1.36) | <0.001 | 1.20 (1.08–1.32) | <0.001 | |
| TNF-alpha | 1.08 (0.98–1.19) | 0.104 | 1.02 (0.91–1.14) | 0.717 | |
| Angptl2 | 1.08 (0.97–1.19) | 0.159 | 1.01 (0.88–1.16) | 0.867 | |
| Cystatin C | 1.15 (1.04–1.26) | 0.005 | 1.32 (1.12–1.55) | 0.001 | |
| Cholinesterase | 0.73 (0.66–0.81) | <0.001 | 0.81 (0.71–0.93) | 0.002 | |

Hazard ratio (95% CI)
0.5   1.0   1.5   2.0

extracellular matrix remodeling, stem cell dysfunction, and cellular senescence and telomere attrition, as shown in model organisms[50]. Unraveling the molecular and genetic underpinnings of slower aging in the extremely old circulatory system may be translatable to the identification of the molecular target for the prevention or delay of cardiovascular diseases in aging.

Inflammation is one of the prominent hallmarks of aging[51] and also of cardiovascular pathology[52]; however, its root causes and pathophysiological roles are diverse in older adults[53]. Our previous study demonstrated that inflammation correlated with physical capability and cognitive function in centenarians and semi-supercentenarians[31], supporting multiple health effects of inflammation at an advanced age. Given a dominance of cardiovascular mortality, proinflammatory cytokines may be less prognostic than circulating NT-proBNP at the highest ages. Angptl2 is an inflammatory mediator specifically associated with an unfavorable metabolic milieu, such as obesity, type 2 diabetes, and hypertriglyceridemia, hence promoting insulin resistance and endothelial inflammation, which are associated with atherosclerosis and heart failure[22,23]. In contrast to its predictability for a range of cardiometabolic outcomes among the general population[54,55], circulating Angptl2 was associated neither with

**Fig. 4 Hazard ratios for death from all causes according to candidate biomarkers.** Hazard ratios (HRs) and 95% confidence intervals (CIs) were calculated with the use of univariate and multivariate Cox proportional hazard models for the entire cohort, those aged 85–99 years at enrollment, 100–104 years at enrollment, and 105 years or older at enrollment. Because the number of individuals aged 110 years or older was too small for multiple analyses, this age group was included in the group over 105 years old. Multivariate analyses were adjusted for the base model covariates; sex, age, educational status, current smoking, history of cardiovascular disease, hypertension, hyperlipidemia, diabetes mellitus, chronic kidney disease (stage 3b-5), CRP (≥0.3 mg/dL), major ECG abnormality, cardiovascular medications, and low plasma albumin (<3.5 g/dL). Each biomarker was entered independently into the models and hazard ratios for each biomarker are reported per 1 SD increment in natural log-transformed values except cystatin C and cholinesterase. For associations of interleukin-6 with mortality, CRP (≥0.3 mg/dL) was excluded from the base model because it is a downstream biomarker of the interleukin-6 pathway[72]. [a] For associations between EC-SOD and mortality, only individuals with 213RR genotype (non-carrier) in *SOD3* (rs1799895) were included in the analysis. A forest plot shows multivariate-adjusted hazard ratio (squares) and 95% confidence interval (horizontal lines). Two-sided *P* values were calculated from Cox proportional hazard models. *NT-proBNP* N-terminal pro-brain natriuretic peptide, *EC-SOD* extracellular superoxide dismutase, *TNF*-alpha tumor necrosis factor-alpha, *Angptl2* angiopoietin-like protein 2.

advancing age nor increased mortality in our oldest old population. As an association between centenarian age and insulin sensitivity has been reported, and centenarian have a very low prevalence of diabetes and obesity, centenarians may exhibit resistance against Angptl2-mediated proinflammatory processes.

Notably, plasma albumin levels were most consistently associated with all-cause mortality in our oldest old cohort. In contrast to age-related increase in prognostic relevance of NT-proBNP, the strength of associations between low albumin and high mortality is markedly stable across age groups. Albumin is an established biomarker of nutritional status[56], but it also related to inflammation[57], hepatic synthesis capacity[58], and increased mortality from cancers[28], respiratory diseases[28], and heart failure[59], suggesting a multifaceted nature of this biomarker. In this study, plasma albumin levels were significantly correlated with cholinesterase levels, inflammation, and even NT-proBNP levels (age- and sex-adjusted partial correlation coefficients: $r = 0.409$, $P < 0.001$; $r = -0.402$, $P < 0.001$; $r = -0.361$, $P < 0.001$; and $r = -0.145$, $P < 0.001$ for cholinesterase, CRP, interleukin-6, and NT-proBNP, respectively, Supplementary Table 3). Collectively, low albumin levels in the oldest old may represent common debilitating processes across a spectrum of ages and pathophysiologies. Despite overall prognostic utility, plasma albumin is less informative with regard to the etiology of mortality.

Our nationwide recruitment of (semi)-supercentenarians lasted over 10 years; hence, a cohort effect must be considered, particularly in light of recent medical innovations. The median birth year of the 123 decedent supercentenarians was 1900. These individuals are less likely to have benefited from potent anti-aging drugs, such as renin-angiotensin system inhibitors and statins than later cohorts, as they were around 81 and 86 years of age when enalapril and pravastatin first appeared on the Japanese market in 1981 and 1986, respectively. Furthermore, the prescription rates for renin-angiotensin system inhibitors were 11.3% and 13.8% in early (born between 1892 and 1899) and later cohorts (born between 1900 and 1906), respectively. The prescription rate of statins was very low among both cohorts (1.9% and 4.6% for early and later cohorts, respectively). Medication utilization is influenced by various factors, such as clinical practice guidelines, and socioeconomic status across countries. In a German study based on the health insurance data of 1,121 unselected centenarians aged ≥100 years in 2013, 55.7% of those living in the community and 42.1% of those in long-term care facilities took renin-angiotensin system inhibitors[60]. The unmedicated supercentenarians studied here, who represent the relatively natural life course of the longest-lived humans, may serve as a valuable index group for future studies of supercentenarians, who are more likely to have received potential anti-aging medications.

There are important limitations to the present study. First, our analytic cohort only represented the Japanese population, limiting the generalizability. Second, there is some heterogeneity between the original three studies from which the entire cohort was aggregated. Obviously, semi-supercentenarians were over-sampled, because the prime aim of this cohort was to discover the biological and genetic basis of supercentenarians an extraordinarily rare phenotype. Although the effect sizes of each biomarker on mortality were generally similar across the age-stratified cohorts, there were differences in directions of effects of some biomarkers (i.e. adiponectin and LDL cholesterol). Therefore, the results of such biomarkers in the total combined cohort should be cautiously interpreted. Third, we obtained participants' medication lists at baseline only, which may not represent life-time exposure to certain medications. In clinical practice, physicians are not without the option of deprescribing anti-hypertensive drugs as their patients get older and frailer[61]. This may at least partly mediate a paradoxical association between hypertension and better survival in the oldest old[62] and even in our own centenarian cohort (Fig. 6e). Therefore, we limited cardiovascular medication to four classes of drug in survival analyses; nitrate, oral anticoagulant, antiarrhythmic drug and digoxin, which are less likely to be withdrawn even in the extremely old. Fourth, not all circulating biomarkers were determined consistently due to variation in the bio-banking of samples. This may render some biomarkers less useful than others. Moreover, our selection of biomarkers was based on the literature regarding cardiovascular morbidity and mortality. It is possible that other untested biomarkers may provide additional prognostic information. Future studies incorporating high through-put technologies are warranted to capture unselected biological pathways underlying extreme longevity. Finally, the causal relationship between low NT-proBNP levels and exceptional survival warrants testing through genetic study, such as Mendelian randomization analysis[63]. We are currently undertaking a genome-wide association study of our centenarian cohorts and plan to report our findings in the near future.

In conclusion, using datasets of the oldest old in Japan, we showed that low levels of circulating NT-proBNP, a potential surrogate for hemodynamic stress, resulting from intrinsic aging in the cardiovascular and renal system, are associated with exceptional survival to the highest ages. These findings identify molecular and pathophysiological pathways that may limit current human longevity. Given the worldwide increase in life expectancy and rising prevalence of older individuals in the total population, understanding the biological effects of aging on major organ systems and the counter-regulatory mechanisms associated with high and exceptional longevity has become a public health priority. Resolving the molecular and genetic basis of the cardioprotective phenotype of supercentenarians may contribute to the identification of therapeutic targets for the prevention or delay of the onset of cardiovascular disease in old age, where cardiovascular and renal aging is likely to play a predominant pathophysiological role.

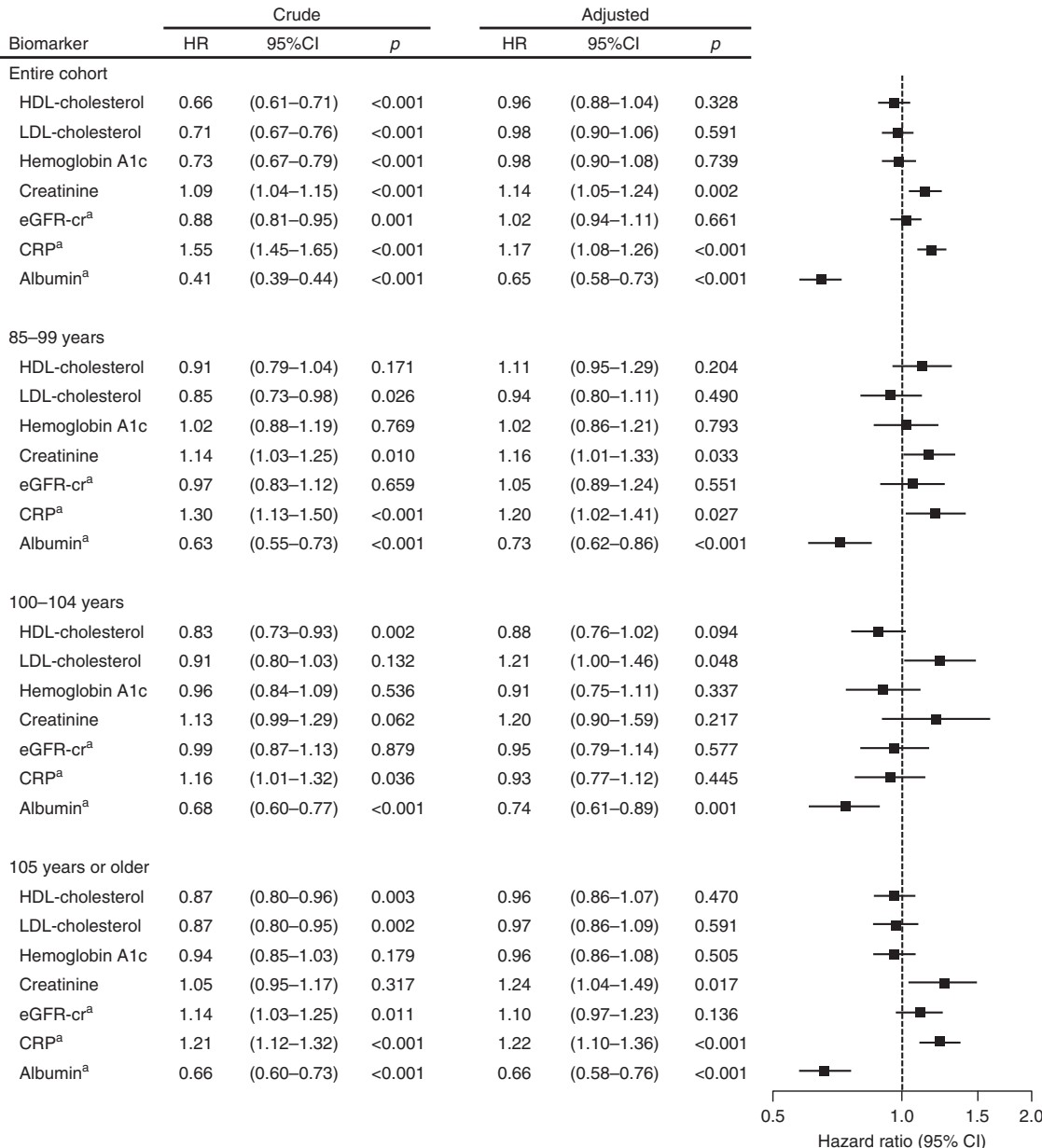

| Biomarker | Crude | | | Adjusted | | |
|---|---|---|---|---|---|---|
| | HR | 95%CI | p | HR | 95%CI | p |
| **Entire cohort** | | | | | | |
| HDL-cholesterol | 0.66 | (0.61–0.71) | <0.001 | 0.96 | (0.88–1.04) | 0.328 |
| LDL-cholesterol | 0.71 | (0.67–0.76) | <0.001 | 0.98 | (0.90–1.06) | 0.591 |
| Hemoglobin A1c | 0.73 | (0.67–0.79) | <0.001 | 0.98 | (0.90–1.08) | 0.739 |
| Creatinine | 1.09 | (1.04–1.15) | <0.001 | 1.14 | (1.05–1.24) | 0.002 |
| eGFR-cr[a] | 0.88 | (0.81–0.95) | 0.001 | 1.02 | (0.94–1.11) | 0.661 |
| CRP[a] | 1.55 | (1.45–1.65) | <0.001 | 1.17 | (1.08–1.26) | <0.001 |
| Albumin[a] | 0.41 | (0.39–0.44) | <0.001 | 0.65 | (0.58–0.73) | <0.001 |
| **85–99 years** | | | | | | |
| HDL-cholesterol | 0.91 | (0.79–1.04) | 0.171 | 1.11 | (0.95–1.29) | 0.204 |
| LDL-cholesterol | 0.85 | (0.73–0.98) | 0.026 | 0.94 | (0.80–1.11) | 0.490 |
| Hemoglobin A1c | 1.02 | (0.88–1.19) | 0.769 | 1.02 | (0.86–1.21) | 0.793 |
| Creatinine | 1.14 | (1.03–1.25) | 0.010 | 1.16 | (1.01–1.33) | 0.033 |
| eGFR-cr[a] | 0.97 | (0.83–1.12) | 0.659 | 1.05 | (0.89–1.24) | 0.551 |
| CRP[a] | 1.30 | (1.13–1.50) | <0.001 | 1.20 | (1.02–1.41) | 0.027 |
| Albumin[a] | 0.63 | (0.55–0.73) | <0.001 | 0.73 | (0.62–0.86) | <0.001 |
| **100–104 years** | | | | | | |
| HDL-cholesterol | 0.83 | (0.73–0.93) | 0.002 | 0.88 | (0.76–1.02) | 0.094 |
| LDL-cholesterol | 0.91 | (0.80–1.03) | 0.132 | 1.21 | (1.00–1.46) | 0.048 |
| Hemoglobin A1c | 0.96 | (0.84–1.09) | 0.536 | 0.91 | (0.75–1.11) | 0.337 |
| Creatinine | 1.13 | (0.99–1.29) | 0.062 | 1.20 | (0.90–1.59) | 0.217 |
| eGFR-cr[a] | 0.99 | (0.87–1.13) | 0.879 | 0.95 | (0.79–1.14) | 0.577 |
| CRP[a] | 1.16 | (1.01–1.32) | 0.036 | 0.93 | (0.77–1.12) | 0.445 |
| Albumin[a] | 0.68 | (0.60–0.77) | <0.001 | 0.74 | (0.61–0.89) | 0.001 |
| **105 years or older** | | | | | | |
| HDL-cholesterol | 0.87 | (0.80–0.96) | 0.003 | 0.96 | (0.86–1.07) | 0.470 |
| LDL-cholesterol | 0.87 | (0.80–0.95) | 0.002 | 0.97 | (0.86–1.09) | 0.591 |
| Hemoglobin A1c | 0.94 | (0.85–1.03) | 0.179 | 0.96 | (0.86–1.08) | 0.505 |
| Creatinine | 1.05 | (0.95–1.17) | 0.317 | 1.24 | (1.04–1.49) | 0.017 |
| eGFR-cr[a] | 1.14 | (1.03–1.25) | 0.011 | 1.10 | (0.97–1.23) | 0.136 |
| CRP[a] | 1.21 | (1.12–1.32) | <0.001 | 1.22 | (1.10–1.36) | <0.001 |
| Albumin[a] | 0.66 | (0.60–0.73) | <0.001 | 0.66 | (0.58–0.76) | <0.001 |

**Fig. 5 Hazard ratios for death from any causes according to traditional cardiovascular risk factors and albumin.** Hazard ratios (HRs) and 95% confidence intervals (CIs) were calculated with the use of univariate and multivariate Cox proportional hazard models for the entire cohort, those aged 85–99 years at enrollment, 100–104 years at enrollment, and 105 years or older at enrollment. Because the number of individuals aged 110 years or older was too small for multiple analyses, this age group was included in the group over 105 years old. Multivariate analyses were adjusted for the base model covariates; sex, age, educational status, current smoking, history of cardiovascular disease, hypertension, hyperlipidemia, diabetes mellitus, chronic kidney disease (stage 3b-5), CRP ( ≥0.3 mg/dL), major ECG abnormality, cardiovascular medications, and low plasma albumin (<3.5 g/dL). Each risk factor (continuous variable) was entered independently into the models and hazard ratios for each risk factor are reported per 1 SD increment except CRP (1 SD increment in natural log-transformed values for CRP). Hyperlipidemia was excluded from the base model for associations of HDL or LDL cholesterol levels with mortality, and diabetes mellitus was excluded for association of hemoglobin A1c with mortality. A forest plot shows multivariate-adjusted hazard ratio (squares) and 95% confidence interval (horizontal lines). Two-sided P values were calculated from Cox proportional hazard models. [a]eGFR-cr, CRP and albumin were entered into the base model as continuous variables instead of categorical variables. HDL indicates high-density lipoprotein, LDL low-density lipoprotein, eGFR-cr estimated glomerular filtration rate based on plasma creatinine.

## Methods

**Study populations**. This study used data from three prospective cohort studies of the oldest old in Japan: the Tokyo Centenarian Study (TCS)[29,30], Japanese Semi-supercentenarian Study (JSS)[14,31], and Tokyo Oldest Old Survey on Total Health (TOOTH)[32]. From the TCS, we identified 1194 eligible individuals aged 100 years or older from the basic residential register of 23 wards of the Tokyo metropolitan area in 2000[29,30]. We sent an invitation letter of our survey to all the eligible centenarians, and 513 participated in our mailed survey, 535 declined, 100 died before recruitment, and 46 had wrong address. Of 513 participants in mailed survey, 304 (65 men, 239 women; mean age, 101.1 ± 1.7 years) participated in the TCS examination by geriatricians at their place of residence between July 2000 and May 2002. Among them, 257 participants (59 men, 198 women, mean age 101.5 ± 1.8 years), of whom plasma sample were available enabling measurements of cardioprotective factors are included in the present study. After initial enrollment, we determined that three of the participants were included in the study at ages of 99.77, 99.73, and 99.93 years, slightly younger than the predefined cutoff age of 100

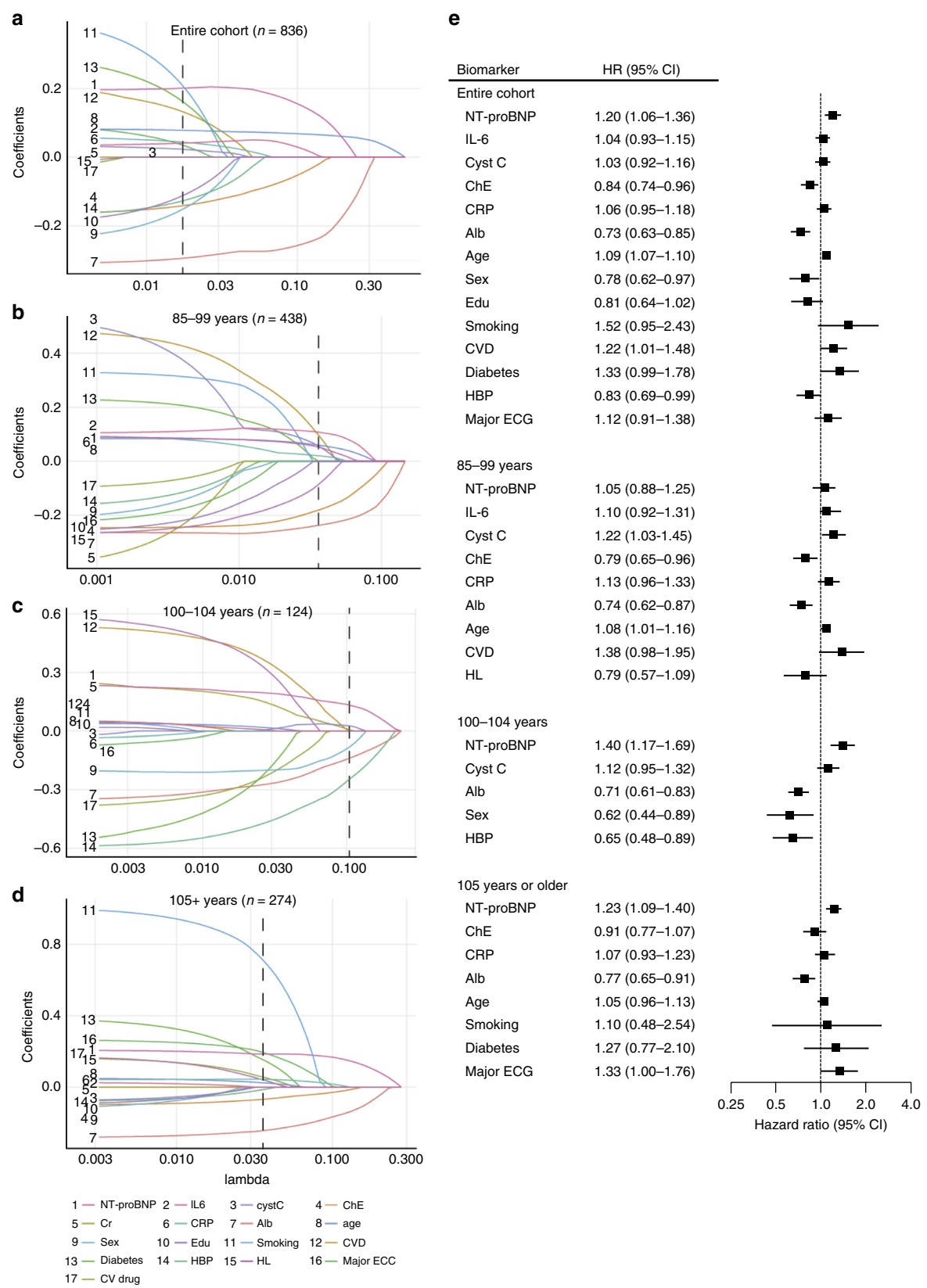

years. However, as these participants had already been evaluated and provided blood samples by the time this error was detected, and the median difference in age at time of enrollment and the study cutoff age was <0.25 years, we decided to include them in our analysis. As with other participants, survival time was calculated from the date of assessment to the day of death (all three participants were confirmed to have died between the ages of 101 and 106 years).

The JSS is a nationwide longitudinal survey that involves mainly semi-supercentenarians, individuals aged 105 years or older[14,31]. Because of extremely low prevalence of semi-supercentenarians (2.0 in 100,000 population according to census 2010), eligible individuals were identified from the centenarians list annually compiled by the Ministry of Health, Welfare and Labour since 1963 until 2002. According to the list, 849 semi-supercentenarians were living in Japan in 2002, including 23 supercentenarians aged 110 years and over. They had been

**Fig. 6 Selected best set of prognostic markers by LASSO-Cox regression analysis.** The least absolute shrinkage and selection operator (LASSO) coefficient profiles of 17 markers associated with mortality were generated for the entire cohort (**a**), those aged 85–99 years at enrollment (**b**), 100–104 years at enrollment (**c**), and 105 years or older at enrollment (**d**). Vertical lines were drawn at the optimal values by using five-fold cross-validation. Lasso coefficients of 17 markers are shown in Supplementary Table 6. **e** Multivariate Cox regression models were performed to calculate hazard ratios (HRs) and 95% confidence intervals (CIs) for LASSO-selected markers (**e**). To identify the best overall set of prognostic markers, the prognostic biomarkers identified in Fig. 4 (NT-proBNP, interleukin-6, cystatin C, and cholinesterase) and Fig. 5 (creatinine, CRP, and albumin) were combined with clinical covariates in the base model; sex, age, educational status, current smoking, history of cardiovascular disease, hypertension, hyperlipidemia, diabetes mellitus, major ECG abnormality, and cardiovascular medications. To standardize the number of participants for the multiple biomarker-risk factor comparisons, we restricted analyses to participants with complete data on all biomarkers. A forest plot shows multivariate-adjusted hazard ratio (squares) and 95% confidence interval (horizontal lines). *NT-proBNP* indicates N-terminal pro-brain natriuretic peptide, *IL-6* interleukin-6, *Cyst C* cystatin C, *ChE* cholinesterase, *CRP* C-reactive protein, *Alb* albumin, *Edu* educational status, *CVD* cardiovascular disease, *HBP* high blood pressure, *HL* hyperlipidemia, *Major ECG* major electrocardiographic abnormality, *CV drug* cardiovascular medication.

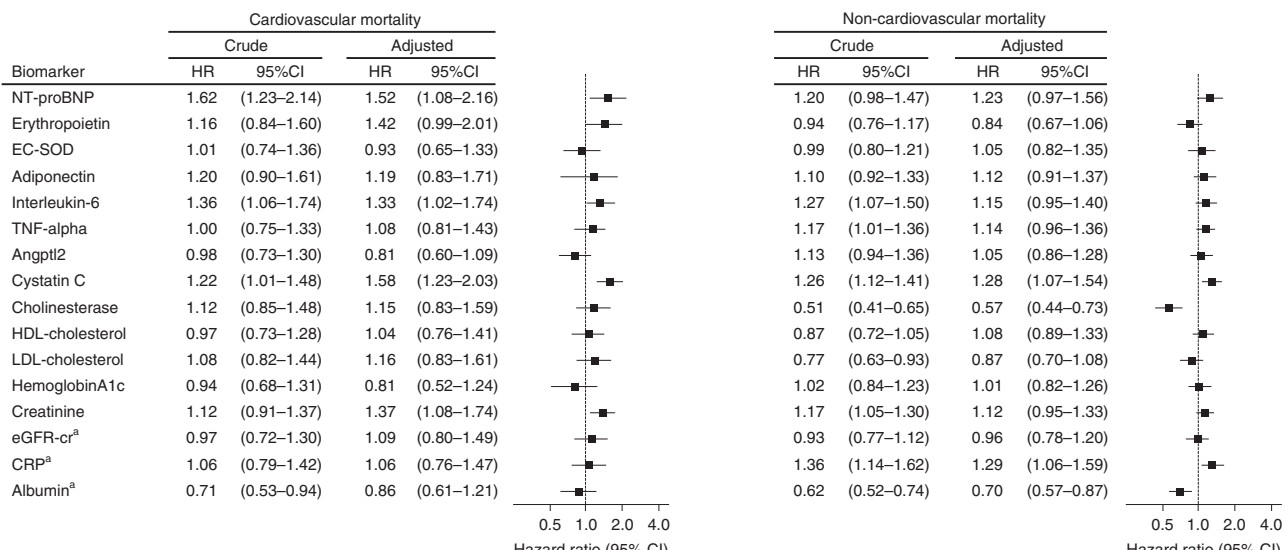

| | Cardiovascular mortality | | | | | | | Non-cardiovascular mortality | | | | |
| | Crude | | Adjusted | | | | | Crude | | Adjusted | | |
| Biomarker | HR | 95%CI | HR | 95%CI | | | | HR | 95%CI | HR | 95%CI |
| NT-proBNP | 1.62 | (1.23–2.14) | 1.52 | (1.08–2.16) | | | | 1.20 | (0.98–1.47) | 1.23 | (0.97–1.56) |
| Erythropoietin | 1.16 | (0.84–1.60) | 1.42 | (0.99–2.01) | | | | 0.94 | (0.76–1.17) | 0.84 | (0.67–1.06) |
| EC-SOD | 1.01 | (0.74–1.36) | 0.93 | (0.65–1.33) | | | | 0.99 | (0.80–1.21) | 1.05 | (0.82–1.35) |
| Adiponectin | 1.20 | (0.90–1.61) | 1.19 | (0.83–1.71) | | | | 1.10 | (0.92–1.33) | 1.12 | (0.91–1.37) |
| Interleukin-6 | 1.36 | (1.06–1.74) | 1.33 | (1.02–1.74) | | | | 1.27 | (1.07–1.50) | 1.15 | (0.95–1.40) |
| TNF-alpha | 1.00 | (0.75–1.33) | 1.08 | (0.81–1.43) | | | | 1.17 | (1.01–1.36) | 1.14 | (0.96–1.36) |
| Angptl2 | 0.98 | (0.73–1.30) | 0.81 | (0.60–1.09) | | | | 1.13 | (0.94–1.36) | 1.05 | (0.86–1.28) |
| Cystatin C | 1.22 | (1.01–1.48) | 1.58 | (1.23–2.03) | | | | 1.26 | (1.12–1.41) | 1.28 | (1.07–1.54) |
| Cholinesterase | 1.12 | (0.85–1.48) | 1.15 | (0.83–1.59) | | | | 0.51 | (0.41–0.65) | 0.57 | (0.44–0.73) |
| HDL-cholesterol | 0.97 | (0.73–1.28) | 1.04 | (0.76–1.41) | | | | 0.87 | (0.72–1.05) | 1.08 | (0.89–1.33) |
| LDL-cholesterol | 1.08 | (0.82–1.44) | 1.16 | (0.83–1.61) | | | | 0.77 | (0.63–0.93) | 0.87 | (0.70–1.08) |
| HemoglobinA1c | 0.94 | (0.68–1.31) | 0.81 | (0.52–1.24) | | | | 1.02 | (0.84–1.23) | 1.01 | (0.82–1.26) |
| Creatinine | 1.12 | (0.91–1.37) | 1.37 | (1.08–1.74) | | | | 1.17 | (1.05–1.30) | 1.12 | (0.95–1.33) |
| eGFR-cr[a] | 0.97 | (0.72–1.30) | 1.09 | (0.80–1.49) | | | | 0.93 | (0.77–1.12) | 0.96 | (0.78–1.20) |
| CRP[a] | 1.06 | (0.79–1.42) | 1.06 | (0.76–1.47) | | | | 1.36 | (1.14–1.62) | 1.29 | (1.06–1.59) |
| Albumin[a] | 0.71 | (0.53–0.94) | 0.86 | (0.61–1.21) | | | | 0.62 | (0.52–0.74) | 0.70 | (0.57–0.87) |

**Fig. 7 Hazard ratios for cardiovascular and non-cardiovascular mortality in the very old according to circulating biomarkers.** Data on cause-specific mortality were available for the very old, aged 85–99 years; 48 died from cardiovascular disease, 41 from cancers, 36 from pneumonia, 5 from advanced dementia, 34 from other causes, and 27 from unknown causes. Those who died from unknown causes were censored in the analysis. Hazard ratios (HRs) and 95% confidence intervals (CIs) for cardiovascular and non-cardiovascular mortality were calculated with the use of univariate and multivariate Cox regression hazard models. Multivariate analyses were adjusted for base model covariates; sex, age, educational status, current smoking, history of cardiovascular disease, hypertension, hyperlipidemia, diabetes mellitus, chronic kidney disease (stage 3b-5), CRP ($\geq$0.3 mg/dL), major ECG abnormality, cardiovascular medications, and low plasma albumin (<3.5g/dL). Hazard ratios for each biomarker are reported per 1 SD increment in natural log-transformed values (from NT-proBNP to Angptl2, and CRP) or per 1 SD (other variables). For associations of interleukin-6 with mortality, CRP ($\geq$0.3 mg/dL) was excluded from the base model because it is a downstream biomarker of the interleukin-6 pathway[72]. Hyperlipidemia was excluded from the base model for associations of HDL or LDL cholesterol levels with mortality, and diabetes mellitus was excluded for association of hemoglobin A1c with mortality. For associations between EC-SOD and mortality, only individuals with 213RR genotype (non-carrier) in *SOD3* (rs1799895) were included in the analysis. [a]eGFR-cr, CRP and albumin were entered into the base model as continuous variables instead of categorical variables. *NT-proBNP* N-terminal pro-brain natriuretic peptide, *EC-SOD* extracellular superoxide dismutase, *TNF-alpha* tumor necrosis factor-alpha, *Angptl2* angiopoietin-like protein 2, *HDL* high-density lipoprotein, *LDL* low-density lipoprotein, *eGFR-cr* estimated glomerular filtration rate based on serum creatinine.

successively recorded on the annual centenarians list since 1997 or earlier. We identified the names and addresses of 543 individuals (82 males and 461 females) among the 849 semi-supercentenarians listed, and sent all of them an invitation letter for a home visit examination. As a result, 135 (115 females and 20 males) participated in the JSS visiting survey. Because the annual centenarian list was discontinued in 2002, our subsequent recruitment of semi-supercentenarians has relied on responses to local governments and nursing homes in the whole country of Japan, or direct inquires by our research team. Between September 2002 and November 2011, a total of 429 centenarians (51 men, 378 women, 90.2% were 105 years or older) were enrolled in the JSS[14]. For the present study, we extended the JSS recruitment until August 2016, which resulted a total of 663 centenarians (79 men, 584 women, 92.2% were 105 years or older). Among them, one individual, who were born before 1890 was excluded because of insufficient age verification, 23 we exclude due to lack of plasma samples, thus a total of 639 centenarians (76 men, 563 women, 91.9% were 105 years or older) were enrolled in the present study. Dates of birth of all participants in the JSS were certified by the national

health insurance or long-term care insurance systems, both of which are linked to the basic residential registration.

During the recruitment process of the TCS and the JSS, we also recruited a first-degree offspring of centenarians ($n = 167$) and their spouses ($n = 167$) as unrelated family[31]. In the Japanese tradition, elder sons are the most likely carers of their parents, thus 75·4% of offspring were male and exactly the same rate were female in the unrelated family cohort. In the present study, unrelated family of centenarians aged between 48 and 94 years (mean age, 73.1 years) were included as younger control for cross-sectional associations of circulating biomarkers with age (Fig. 2). Present and past diseases of this group were as follows; hypertension (41.4%), diabetes mellitus (14.2%), hyperlipidemia (20.4%), stroke (3.6%), and coronary heart disease (5.4%). Classification of cardiovascular abnormality in unrelated families of centenarians was based on medical history and medication list because of lack of ECG assessment in this population.

The TOOTH survey is a community-based prospective cohort study of the very old[32]. Between March 2008 and November 2009, we identified 2,875 eligible

**Table 2 The prognostic performance for all-cause mortality in the entire and age-stratified cohort based on Cox proportional hazard models.**

| | C-index | 95 % CI | P-value[a] | Optimism[b] | Optimism-corrected C-index[c] |
|---|---|---|---|---|---|
| **(a) Entire cohort** (n = 838) | | | | | |
| Base Model | 0.778 | 0.759–0.796 | – | 0.006 | 0.772 |
| Base Model + NT-proBNP | 0.786 | 0.768–0.804 | <0.001 | 0.006 | 0.781 |
| Base Model + Interleukin-6 | 0.781 | 0.763–0.799 | 0.001 | 0.006 | 0.775 |
| Base Model + Cystatin C | 0.781 | 0.764–0.799 | 0.004 | 0.006 | 0.775 |
| Base Model + Cholinesterase | 0.784 | 0.766–0.802 | 0.002 | 0.005 | 0.779 |
| Base Model + All biomarkers | 0.791 | 0.773–0.809 | <0.001 | 0.007 | 0.784 |
| **(b) 85–99 years** (n = 438) | | | | | |
| Base Model | 0.648 | 0.602–0.695 | – | 0.035 | 0.614 |
| Base Model + NT-proBNP | 0.667 | 0.621–0.714 | 0.030 | 0.036 | 0.632 |
| Base Model + Interleukin-6 | 0.656 | 0.609–0.702 | 0.117 | 0.037 | 0.619 |
| Base Model + Cystatin C | 0.655 | 0.609–0.701 | 0.195 | 0.035 | 0.621 |
| Base Model + Cholinesterase | 0.669 | 0.623–0.716 | 0.041 | 0.033 | 0.636 |
| Base Model + All biomarkers | 0.681 | 0.635–0.727 | 0.017 | 0.038 | 0.643 |
| **(c) 100–104 years** (n = 126) | | | | | |
| Base Model | 0.645 | 0.591–0.700 | – | 0.050 | 0.595 |
| Base Model + NT-proBNP | 0.657 | 0.603–0.711 | 0.133 | 0.051 | 0.606 |
| Base Model + Interleukin-6 | 0.647 | 0.594–0.701 | 0.268 | 0.053 | 0.594 |
| Base Model + Cystatin C | 0.652 | 0.598–0.706 | 0.211 | 0.053 | 0.599 |
| Base Model + Cholinesterase | 0.644 | 0.590–0.698 | 0.987 | 0.054 | 0.590 |
| Base Model + All biomarkers | 0.660 | 0.607–0.714 | 0.079 | 0.061 | 0.599 |
| **(d) 105 years or older** (n = 274) | | | | | |
| Base Model | 0.617 | 0.577–0.656 | – | 0.029 | 0.588 |
| Base Model + NT-proBNP | 0.653 | 0.615–0.691 | 0.001 | 0.027 | 0.625 |
| Base Model + Interleukin-6 | 0.626 | 0.586–0.666 | 0.058 | 0.030 | 0.596 |
| Base Model + Cystatin C | 0.620 | 0.581–0.658 | 0.312 | 0.031 | 0.589 |
| Base Model + Cholinesterase | 0.636 | 0.596–0.676 | 0.019 | 0.027 | 0.609 |
| Base Model + All biomarkers | 0.664 | 0.626–0.703 | <0.001 | 0.032 | 0.632 |

The base model includes sex, age, educational status, current smoking, history of cardiovascular disease, diabetes mellitus, hypertension, hyperlipidemia, chronic kidney disease (stage 3b-5), elevated CRP (≥0.3 mg/dL), major ECG abnormality, cardiovascular medications, and low plasma albumin (<3.5 g/dL). To standardize the number of participants for the multiple biomarker comparisons, we restricted analyses to participants with complete data on all biomarkers.
NT-proBNP N-terminal pro-brain natriuretic peptide, C-index concordance index, CI confidence interval.
[a]One-sided-P values for the C-index are computed by assuming asymptotic normality.
[b]Optimism was calculated by bootstrap method with 100,000 resampling to obtain an accurate value to the third decimal place.
[c]Optimism-corrected C-index was calculated as the optimism is subtracted from the C-index in the original model.

individuals born before January 1923 and living in the community of the Tokyo metropolitan area from the basic registry of residents of the Tokyo Metropolitan area. Of these, 447 were unable to contact, 663 declined, 403 were refused by family or care giver, 210 were withheld because of poor health conditions (i.e. acute illness, severe dementia), thus a total of 1152 were recruited, of which 168 participated in self/proxy-completed questionnaire only, and 984 participated in face-to face interview. Among the respondent of face-to face interview, 542 (236 men, 306 women) participated in the TOOTH medical and dental examination. Of these, nine individuals lacking plasma sample and two aged 100 years or older at enrollment were excluded, thus 531 subjects (233 men, 298 women, mean age 87.8 ± 2.0 years old) were enrolled in this analysis.

We have complied with all relevant regulations for work with human subjects. Written informed consent to participate in the present study was obtained either from the participants or their proxy when the individual lacked the capacity to consent. All cohort studies were approved by the ethics committee of the Keio University School of Medicine (ID: 20021020, 20022020, 20070047), and are registered in the University Hospital Medical Information Network Clinical Trial Registry as observational studies (ID: UMIN000040446, UMIN000040447, UMIN000001842).

**Baseline examination.** All participants were examined directly by experienced geriatricians (N.H., Y.A., and M.T.) at the time of enrollment, in accordance with previously described protocols[14,29–32]. Our assessment protocols include socio-economic status, previous medical histories and present medical conditions, current medication use, and various lifestyle factors including smoking and alcohol drinking, and physical and cognitive function. In assessment, medical interview and physical examination including heart sound, respiratory sound and presence of legs edema were conducted by trained geriatricians to capture the clinical signs of congestion. Systolic and diastolic blood pressure (BP) and heart rate were measured twice using an automatic sphygmomanometer in a seated position. We used the average of the two measurements for the analysis. We categorized the present medical condition based on the International Classification of Diseases, 10th Revision categories as follows; Hypertension [I10], Coronary Heart Disease [I20–I25], Stroke [I60–I69], Diabetes mellitus [E11], Hyperlipidemia [E78].

Because of the high percentage of undiagnosed diabetes mellitus in the oldest old population, we defined diabetes as fulfilling one or more criteria: (1) self-reported diagnosis, (2) administration of insulin or other oral hypoglycemic medications, (3) random plasma glucose ≥200 mg/dL, or (4) hemoglobin A1c (HbA1c) ≥6.5%. Hyperlipidemia was defined as a low-density lipoprotein cholesterol (LDL-C) level ≥140 mg/dL, or triglyceride levels ≥200 mg/dL, or current use of medication for dyslipidemia. Hypertension was defined as current use of medication for hypertension or self-reported diagnosis. Anemia was defined as hemoglobin levels <12.0 g/dL for women or <13.0 g/dL for men, respectively according to WHO criteria for anemia for adults. Basic activities of daily living (ADL) were assessed using the Barthel Index, and cognitive function was evaluated according to the Mini-Mental State Examination (MMSE). Because approximately 20% of centenarians could not complete the MMSE due to visual/hearing impairment or an inability to communicate, they were simultaneously evaluated using the Clinical Dementia Rating (CDR) scale in participants of the TCS and the JSS.

We recorded standard 12-lead ECG with each participant in the supine position using standardized procedures. ECG findings were coded by two cardiology specialists (S.Y. and A.K.) blinded to the participants' clinical information using the Minnesota codes. Participants were considered to have a cardiovascular abnormality when one or more of the following criteria were fulfilled: (1) a history of coronary heart disease or stroke, (2) cardiovascular medication use (i.e., nitrate, oral anticoagulant, antiarrhythmic drug, or digoxin), and (3) a major ECG abnormality, including old myocardial infarction, pacemaker rhythm, atrial fibrillation or flutter, left ventricular hypertrophy, advanced atrioventricular block, left bundle branch block, and Wolff–Parkinson–White syndrome. Those who reported pacemaker implantation but lacked ECG documentation (n = 2 in the very old, n = 2 in centenarians aged 100–104 years, and n = 4 in semi-supercentenarians aged 105–109 years, respectively) were considered to have a cardiovascular abnormality.

**Circulating biomarkers.** Non-fasting blood samples were collected from all participants at the time of enrollment, and stored at −80 degree until subsequent analysis. Plasma levels of NT-proBNP were measured using electro-chemiluminescence immunoassay (ECLIA) method, and erythropoietin was

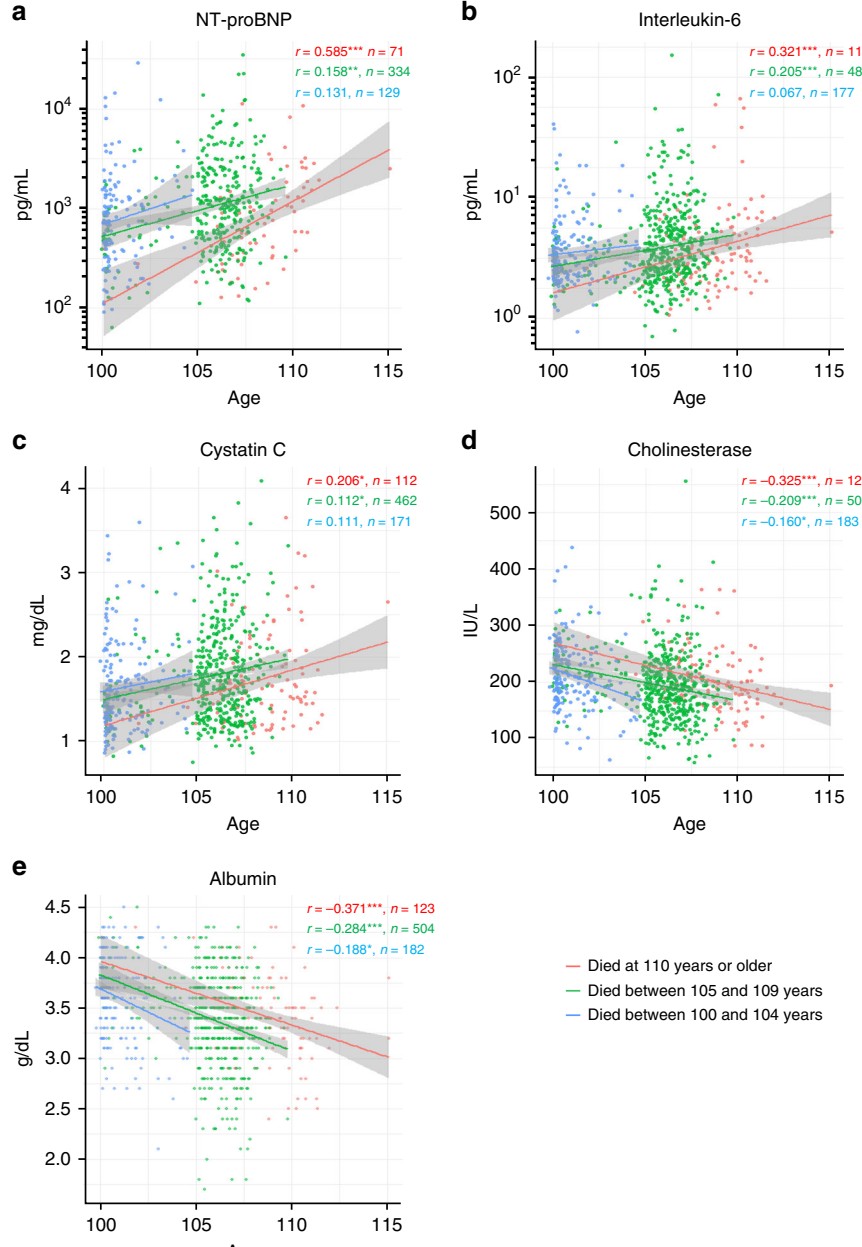

**Fig. 8 Prognostic biomarkers in centenarians stratified by the age at death.** Spearman's correlation coefficients of the relationships between prognostic biomarkers identified in Fig. 4 and albumin, and age at enrollment were calculated within decedent centenarian categories: decedent centenarians (died between 100–104 years, blue line), decedent semi-supercentenarians (died between 105–109 years, green line), and decedent supercentenarians (died at ≥110 years, red line). The shaded area represents the 95% confidence interval of the correlation line. All the biomarkers were assessed at the time of enrollment. Population sizes for the five biomarkers differ due to variation in the bio-banking of samples. *NT-proBNP* indicates N-terminal pro-brain natriuretic peptide. *p < 0.05, **p < 0.01, ***p < 0.001.

measured using chemiluminescent enzyme immunoassay (CLEIA) method (SRL Inc., Tokyo, Japan). Inter-assay coefficient of variations of NT-porBNP and erythropoietin were 1.57%, and 3.77%, respectively. High-molecular weight (HMW) adiponectin concentrations were measured in duplicate using commercially available ELISA kits (Otsuka pharmaceutical Co., Ltd. Tokyo, Japan). The inter-assay coefficient of variance of HMW-adiponectin was 11.0%. Plasma EC-SOD concentrations were measured by enzyme-linked immunosorbent assay (ELISA) method as previously described[64]. The inter-assay coefficient of variance was 6.5%.

Plasma levels of interleukin-6, and TNF-alpha were measured in duplicate using commercially available ELISA kits [Quantikine HS (Human IL-6), R&D Systems, Minneapolis, U.S.A; Quantikine HS (Human TNF-alpha), R&D Systems, Minneapolis, U.S.A; respectively]. Inter-assay coefficient of variations of interleukin-6 and TNF-alpha were 9.43%, and 8.72%, respectively. Plasma Angptl2 levels were measured in duplicate by using the human Angptl2 ELISA as previously described[22]. The intra-assay and inter-assay coefficients of variance were 1.65% and

3.13%, respectively. Plasma cystatin C was measured using colloidal gold immunoassay method, and the intra-assay and inter-assay coefficients of variance were 1.72% and 0.79%, respectively (SRL Inc., Tokyo, Japan). Plasma cholinesterase activity was measured with the Japan Society of Clinical Chemistry (JSCC) proposed reference method, and the intra-assay coefficients of variance were 1.08%, and Plasma albumin was measured using the bromocresol purple (BCP) method, and hemoglobin concentrations were measured using automated method (SRL Inc., Tokyo, Japan). We measured plasma total cholesterol (TC), triglyceride (TG), and high-density lipoprotein cholesterol (HDL-C) levels using standard enzymatic methods. LDL-C was calculated using Friedewald's formula when the TG level was <400 mg/dL. We measured plasma glucose levels using hexokinase method. The glycated hemoglobin (HbA1c) level was determined using high-performance liquid chromatography (HPLC) method and reported as the National Glycohemoglobin Standardization Program (NGSP) values. The serum creatinine level was measured using a standard enzymatic method, and the estimated

glomerular filtration rate (eGFR) by serum creatinine was calculated according to Clinical Practice Guidebook for Diagnosis and Treatment of Chronic Kidney Disease 2012 (Japanese Society of Nephrology): eGFR (mL/min/1.73 m$^2$) = 194* [serum creatinine (mg/dL)]$^{-1.094}$*[age (years)]$^{-0.287}$[*0.739 (if female)][65].

**Genotyping of Arg213Gly polymorphism at SOD3 locus (rs1799895).** We genotyped a common missense mutation in codon 213 in exon 3 of SOD3 (rs1799895) as previously described[33]. Total DNA was extracted from whole blood by using the FlexGene DNA Kit (QIAGEN, Hilden, Germany) and was stored in FG3 solution at 4 °C. The Arg213Gly Polymorphism at SOD3 locus was genotyped with TaqMan prove.

**Study outcomes.** The primary outcome was all-cause mortality. Participants in both the TCS and JSS were followed-up annually until October 2016 via telephone contact or a mail survey, resulting in a 251,339 person-day follow-up (median period, 740 days; range, 3–5420 days) for the TCS and a 354,856 person-day follow-up (median period, 411 days; range, 1–2682 days) for the JSS. During follow-up, 5 (1.9%) participants of the TCS, and 16 (2.5%) participants of the JSS were lost to follow-up. Participants in the TOOTH survey were followed-up for six years until December 2015 via annual telephone contact or a mail survey. For the TOOTH study, information on the cause of death (i.e., cardiovascular disease, cancer, pneumonia, severe dementia, or other) was obtained by telephone contact with the family or a proxy. Follow-up in the TOOTH study was continued until one of the following censoring events occurred: loss to follow-up (e.g., moving away from the Tokyo Metropolitan area, $n = 65$), or the end of the 6-year follow-up period.

**Statistical analysis.** Baseline characteristics are expressed as means and standard deviations (SD) or medians and interquartile ranges; categorical variables are shown as numbers and proportions. Skewed variables were logarithmically transformed on a natural log scale. Participants were classified into four groups based on the age at enrollment (85–99 years, 100–104 years, 105–109 years, and ≥110 years). We analyzed trends in each parameter across age groups using the trend test for continuous variables, and the Cochran-Armitage trend test for categorical variables. Spearman's rank correlation coefficients and 95% confidence intervals (95% CIs) were calculated to identify associations between age at enrollment and each biomarker according to presence or absence of a cardiovascular abnormality. In addition, multivariable stepwise linear regression with backward elimination was constructed to examine the independent association between baseline characteristics and circulating biomarkers.

Associations between biomarkers and all-cause mortality were assessed using univariable and multivariable Cox proportional hazard models both for the total combined cohort and age-specific groups (85–99 years, 100–104 years, and ≥105 years). Base models were constructed with sequential adjustment for demographic and lifestyle factors (sex, age at baseline [as a continuous variable], educational status [high school degree or higher], and current smoker), followed by traditional cardiovascular risk factors (history of cardiovascular disease, hypertension, hyperlipidemia, diabetes mellitus, chronic kidney disease (stage 3b–5)[66], elevated CRP (≥0.3 mg/dL)[67], major ECG abnormality, and cardiovascular medication [nitrate, oral anticoagulant, antiarrhythmic drug, and digoxin]), and then further adjustment for low albumin levels (<3.5 g/dL)[28]. The prespecified cutoff points for chronic kidney disease, CRP, and albumin were adopted from epidemiological literature on the risk of cardiovascular and all-cause mortality[28,66,67]. Subsequently, each biomarker was entered independently to the base model as both a continuous (per 1 SD increment) and categorical variable (using cutoff points). When established cutoff points for the biomarkers were lacking, we used age-group-specific tertiles for each biomarker. Finally, to identify the best overall set of predictors, all the prognostic biomarkers significantly associated with mortality in the multivariate analysis were combined with clinical covariates in the base model. We employed three different techniques to select the most useful prognostic markers in the final model; (1) the least absolute shrinkage and selection operator for Cox regression (LASSO-Cox) with five-fold cross-validation, (2) a stepwise forward selection with inclusion criteria $P < 0.20$, and (3) forced entry models. The LASSO is a penalized technique for variable selection that is effective when the number of events per variables is low[68,69]. LASSO shrinks coefficients for weaker predictors toward zero. The degree of shrinkage is determined by an optimal parameter lambda, as identified by five-fold cross-validation. To standardize the number of participants for the multiple biomarker-risk factor comparisons, we restricted subsequent analysis to participants with complete data on all biomarkers being studied for all-cause mortality.

To assess the prognostic performance of each biomarker, we computed Concordance index (C-index); we compared the base model with the models that incorporated each biomarker with the base model by calculating C-index based on Cox proportional hazard models[34]. P values for the C-index are computed by assuming asymptotic normality. To correct the overfitting of the statistical models in relatively small dataset[70], we estimated the optimism of the developed models according to Harrell et al.[71], where the estimated optimism was calculated as naïve C-index for the bootstrapped samples minus C-index evaluated on the original dataset. This process was repeated 100,000 times and averaged to gain an accurate estimate of the optimism to the third decimal place. The optimism was then subtracted from the C-index of the original model to provide optimism-corrected C-index.

Additionally, we performed a retrospective analysis examining the associations between each biomarker and age at enrollment across deceased centenarian categories (age at death: 100–104 years, 105–109 years, and ≥110 years), using Spearman's correlation analysis.

Analyses were performed using STATA SE 13 software (Stata Corp LP, College Station, TX, USA), and R (version 3.4.3. and glmnet package for LASSO selection; version 3.6.1. and survcomp package (version 1.36.1) for calculating C-index and optimisms). All P values were two-tailed except for the C-index based on Cox hazard models, and a P value < 0.05 was considered statistically significant.

**Reporting summary.** Further information on research design is available in the Nature Research Reporting Summary linked to this article.

## Data availability
The datasets analyzed in the current study have ethical and legal restrictions for public deposition due to the inclusion of sensitive information from a vulnerable population. The data will be available upon request with an appropriate research arrangement with approval of the Research Ethics Committee of Keio University School of Medicine for Clinical Research. Thus, to request the data, please contact Dr. Yasumichi Arai (corresponding author) via e-mail: yasumich@keio.jp

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

## Acknowledgements

The authors thank the participants; Ms. Miho Shimura for assistance in participants' recruitment. The list of investigators who participated in the TCS, JSS, or TOOTH but did not meet the author's criteria is shown in Supplementary Note 1. This study was supported by a grant from the Ministry of Health, Welfare, and Labour for the Scientific Research Project for Longevity; a Grant-in-Aid for Scientific Research (No 23617024, 21590775, 15KT0009) from the Japan Society for the Promotion of Science; the Program for Initiative Research Projects from Keio University and Keio University Global

Research Institute (KGRI), the Program for an Integrated Database of Clinical and Genomic Information (JP16kk0205009) and the Platform Program for Promotion of Genome Medicine (JP17km0405103) from Japan Agency for Medical Research and Development; and the Medical-Welfare-Food-Agriculture Collaborative Consortium Project from the Japan Ministry of Agriculture, Forestry, and Fisheries.

## Author contributions

Y.A., S.Y., and N.H. conceived the study design. N.H., Y.A., M.T., Y.Abe., T.S., H.I., M.E., J.M., Y.O., and T.A. participated in data collection. T.H., Y.A., K.Y., S.Y., A.K., Y.Abe., T.S., H.I., M.E., J.M., and T.T. participated in data analysis and interpretation. T.H., K.Y., and T.S. did the final statistical analysis. Y.Abe assisted with data preparation. Y.O., T.A., H.O. provided critical revision of the draft. T.H., K.Y., and Y.A. drafted the report. All authors approved the final version of the report.

## Competing interests

The authors declare no competing interests.
