## [Peer Review File · Nature Communications]

Reviewers' comments:

Reviewer #1 (Remarks to the Author):

Summary

This study of 531 subjects (233 men, 298 women, mean age, 87.8 ± 2.0 years) was based on three prospective cohort studies of the oldest old in Japan; the Tokyo Centenarian Study (TCS), the Japanese Semi-supercentenarian Study (JSS), and the Tokyo Oldest Old Survey on Total Health (TOOTH).

The main goal of the study was to identify cardioprotective pathways associated with extreme longevity. The independent variables included plasma levels to N-terminal pro-brain natriuretic peptide (NT-proBNP), erythropoietin, adiponectin, and extracellular superoxide dismutase (EC-SOD or SOD3). Plasma concentrations of four inflammatory biomarkers: C-reactive protein (CRP), interleukin-6, tumor necrosis factor-alpha (TNF-alpha), and angiotensin-converting enzyme 2 (Angptl2) were also included in the analyses.

The primary outcome for entire study cohort was all-cause mortality.

The median follow-up period for the TCS was 740 days and 411 days for the JSS. Participants in the TOOTH survey were followed up for six years.

In additional analyses the probability of all-cause mortality according to presence or absence of cardiovascular abnormality separately for three age group (85–99years, 100–104 years, and 105 years or older).

During the follow-up periods, 70.1% of the study participants died: 191 in the very old cohort (7.8 per 100 person-years), 270 in centenarians aged 100–104 years at baseline, 506 deaths among semi-supercentenarians and 33 deaths among supercentenarians.

A retrospective analysis was also conducted in centenarians based on age at death of 1,427 oldest-old individuals and examined associations between each biomarker and age at enrollment across deceased centenarian categories, including 36 supercentenarians (aged ≥ 110 years), 572 semi-supercentenarians (105–109 years), 288 centenarians (100–104 years), and 531 very old persons (85–99 years) at enrollment.

The results linked molecular pathways linked to concomitant increases in endogenous cardioprotective factors and inflammatory mediators, accompanied with reductions in major organ reserves to both extended longevity and manifestation of heart failure, Despite these indicators, the longest-lived individuals, 56% of whom were not taking CV medication remained asymptomatic. Four prognostic molecular predictors of all-cause mortality among the oldest old, that are independent of traditional risk factors were identified: NT-proBNP, interleukin-6, cystatin-C, and cholinesterase. Notable among these biomarkers is NT-proBNP, which plays a distinct role in cardiovascular diagnosis, and predicts exceptional survival to supercentenarian age.

The authors conclude 1) that low production of NT-proBNP is a key biological pathway of exceptional survival into the highest ages and intrinsic aging in the circulatory system ultimately results in the deterioration of hemodynamic homeostasis, which eventually limits survival at the current ceiling of human longevity; and 2) that retardation of age-related increases in NT-proBNP in addition to traditional risk factors for clinically prevalent cardiovascular disease merits consideration as a potential therapy to enhance longevity. Finally, the authors suggest unmedicated supercentenarians studied here, who represent the relatively natural life course of the longest lived humans, may serve as a valuable index group for future studies of supercentenarian, who are more likely to have received potential anti-aging medications.

Comments

The study was well designed and data have been thoroughly analyzed and clearly reported

Given the world-wide prevalence of older persons of that major increases will occur in the prevalence of every person, it is imperative that we elucidate factors that are associated with high and extreme longevity that go beyond risk factors for diseases that occur more often in older persons.

Aging is the major risk factor for the most deadly clinical diseases and becomes manifested as time-dependent in our organ reserve functions. In other words aging extracts chaos from ordered behavior within our body cells and their interactions with other matrix and with other cells. I would encourage the authors to address even a broader issue in the discussion of their paper. It's aging (time of our existence) itself, not a disease. The effects of time are numerous. Surely, aging may be construed as a slowly progressive disease that inevitably results in death, often without the occurrence of a clinical disease diagnosis. Extreme longevity or early mortality conferred to some on the basis of genetics, life style and environmental effects modulate the effect at which the rate of time-dependent deterioration that we refer to as aging. In this context, a disease can be construed as exaggerated effects of time or accelerated aging.

Reviewer #2 (Remarks to the Author):

Associations of cardiovascular biomarkers with exceptional survival to the highest ages. Hirata et al.

This paper describes the investigation of a unique line up of three Japanese cohorts comprising very old people (85-99 years N=531), centenarians (100-104 years N=288), semi-supercentenarians 105-109 years N=572) and supercentenarians (>110 years N=36). The study is focused on the question whether a set of cardiovascular disease markers associate to survival to the highest ages.

To this end, a battery of 9 cardiovascular biomarkers in the circulation were measured in the 1,427 long-lived people and cross-sectional analyses of age strata, as well as prospective analyses on all cause mortality were performed. The authors concluded that NTproBNP levels in long-lived people of all age groups are associated with and predictive of all-cause mortality. Adding these markers to existing clinical data does not add much to mortality prediction.

Except for the uniqueness of the study population, it is not so clear what exactly the novelty is in this paper that warrants publication in Nat Commun. NTproBNP is a known predictor of mortality in coronary artery disease, and the current paper adds that this now also seems the case in extremely long-lived people. Given the many data that were so carefully collected in this nice study I would like to urge the authors to highlight much better what the most interesting novelty is of this paper. The paper is presented almost as a search for the best predictive marker that in the end does not predict

too much better than classical clinical variable (which is also truly hard to do, most novel markers cannot do that). The paper seems solid in methodology and epidemiological sense. There may be much more to gain in aetiological sense I think.

Some info on the choice of biomarkers and previous work explaining why these were selected (based on previous studies of CVD risk and mortality should be added to the introduction/discussion). The introduction states that CVD risk in centenarians is low so endogenous counter regulation mechanism against CVD are expected to be found among the centenarians. The paper focuses on markers of CVD risk stating that it is known that lipid and IR risk factors are low in the centenarians, although ECG abnormalities have been found. The choice of markers is only explained in experimental procedures. It should be stated in the intro why the markers were selected : indicate heart failure better than other markers ? Especially indicates ECG abnormalities ? The materials and methods by the way explains that four biomarkers were selected, but actually 9 were investigated. And it was indicated that centenarians are low in lipid and IR risk factors, so one would like to see whether the mortality is predicted better than these, but I could not find lipid or glucose related markers in table 1. Hyperglycaemia and hyperlipidemia are in the base model, but these are the thresholds for diagnosis.

The paper is focused on CVD risk but not only on CVD mortality. Is the main cvd mortality death cause heart failure ? Do the chosen markers not indicate risk for other types of death causes ? if death causes are not available then what about other studies ? Is in the four age categories the prevalence of disease and death causes known. The medical conditions at baseline were recorded. Which of these conditions was associated with mortality ? Which of the markers tested was associated with these conditions ? Figure 2 shows association of baseline markers with age for individuals with or without the cardiovascular conditions (three criteria were mentioned: CVD disease history, medication use and ECG abnormalities) . One would like to know which of these predict mortality best (and is that so in each of the age categories) and is that prediction better than that of non CVD medical baseline conditions. And with respect to the markers one would like to know with which relevant medical condition there is baseline association other than CVD conditions. A side remark following the discussion: renal function and haemodynamic stress is known to be so relevant in elderly, why is creatinine or the use of diuretics than not among the markers.

With respect to Figure 2: The controls as I understand are mainly women (the wives of the oldest son dominating the offspring population). Is the correlation the same if one confines the correlations only to women (the oldest categories mainly consist of women but the very old not).

The authors in their abstract claim that the findings expand biological knowledge. Burt the paper is focused on testing biomarkers in epidemiological sense (which predicts mortality best).

Please, describe in the discussion not only that there are associations identified, but also in which direction the associations have been found. F.e. in page 8 line215: ..., these results suggest that circulating NT-proBNRP is an important biological correlate of exceptional survival to the highest ages => indicate whether this is about low or high levels. Please check the document for the description of direction of effects.

Low NT-proBNP seems to associate with extreme survival: could the authors speculate in the discussion whether (super)centenarians are reflecting a selection of people with low lifetime risk of

cardiovascular disease? Is the working hypothesis now that (super)centenarians have delayed their cardiovascular risk by 30 years? What about other disease and death risks? The discussion section is not so clear yet in the proposed hypothesis.

The question is whether in aetiological sense one can show that neurohormonal activation (indicated by NT-proBNP) is more relevant for survival than inflammation, lipid metabolism, organ reserve, reflected by the other markers. Why was only the SOD3 variant investigated. If (super)centenarian seem to have a history of low levels of NT-proBNP, it would have been logical to test for the prevalence of SNPs contributing to NT-proBNP levels. rs198389 and rs61761991 in the gene NPPB have previously been associated: are the genotype distributions of these SNPs changing over the age strata? Could one test the causality of the NT-proBNP level to longevity using these SNPs (Mendelian Randomization). There are variants for IL6 (even associated with longevity in previous studies), adiponectin etc.

The authors in their abstract claim that the findings may have public health implications. Which?

ID: NCOMMS 18-33291A

Title: Associations of Cardiovascular Biomarkers with Exceptional Survival to the Highest Ages

Authors: Hirata T, Ara Y, Yuasa S, Abe Y, Takayama M, Sasaki T, Kunitomi A, Inagaki H, Endo M, Morinaga J, Adachi T, Oike Y, Takebayashi T, Okano H, and Hirose N.

Response to Reviewers' comment and manuscript changes

We appreciate the careful review and constructive suggestions by the two reviewers. Based on their comments, we have reanalyzed and reviewed our dataset and thoroughly discussed the biological as well as clinical factors that contribute to exceptional survival. We are confident that the manuscript has been substantially improved and provides unique insights into the effects of cardiovascular aging on human longevity. Original reviewer comments in italics typeface, and responses in regular typeface.

Reviewer #1 (Remarks to the Author):

Summary

This study of 531 subjects (233 men, 298 women, mean age, 87.8 ± 2.0 years) was based on three prospective cohort studies of the oldest old in Japan; the Tokyo Centenarian Study (TCS), the Japanese Semi-supercentenarian Study (JSS), and the Tokyo Oldest Old Survey on Total Health (TOOTH).

The main goal of the study was to identify cardioprotective pathways associated with extreme longevity. The independent variables included plasma levels to N-terminal pro-brain natriuretic peptide (NT-proBNP), erythropoietin, adiponectin, and extracellular superoxide dismutase (EC-SOD or SOD3. Plasma concentrations of four inflammatory biomarkers: C-reactive protein (CRP), interleukin-6, tumor necrosis factor-alpha (TNF-alpha), and angiotensin-converting enzyme 2 (Angptl2) were also included in the analyses.

The primary outcome for entire study cohort was all-cause mortality.

The median follow-up period for the TCS was 740 days and 411 days for the JSS.

Participants in the TOOTH survey were followed up for six years.

In additional analyses the probability of all-cause mortality according to presence or absence of cardiovascular abnormality separately for three age group (85–99years, 100–104 years, and 105 years or older.

During the follow-up periods, 70.1% of the study participants died: 191 in the very old cohort (7.8 per 100 person-years), 270 in centenarians aged 100–104 years at baseline, 506 deaths among semi-supercentenarians and 33 deaths among supercentenarians.

A retrospective analysis was also conducted in centenarians based on age at death of 1,427 oldest-old individuals and examined associations between each biomarker and age at enrollment across deceased centenarian categories, including 36 supercentenarians (aged ≥ 110 years), 572 semi-supercentenarians (105–109 years), 288 centenarians (100–104 years), and 531 very old persons (85–99 years) at enrollment.

The results linked molecular pathways linked to concomitant increases in endogenous cardioprotective factors and inflammatory mediators, accompanied with reductions in major organ reserves to both extended longevity and manifestation of heart failure, Despite these indicators, the longest-lived individuals, 56% of whom were not taking CV medication remained asymptomatic. Four prognostic molecular predictors of all-cause mortality among the oldest old, that are independent of traditional risk factors were identified: NT-proBNP, interleukin-6, cystatin-C, and cholinesterase. Notable among these biomarkers is NT-proBNP, which plays a distinct role in cardiovascular diagnosis, and predicts exceptional survival to supercentenarian age.

The authors conclude 1) that low production of NT-proBNP is a key biological pathway of exceptional survival into the highest ages and intrinsic aging in the circulatory system ultimately results in the deterioration of hemodynamic homeostasis, which eventually limits survival at the current ceiling of human longevity; and 2) that retardation of age-related increases in NT-proBNP in addition to traditional risk factors for clinically prevalent cardiovascular disease merits consideration as a potential therapy to enhance longevity. Finally, the authors suggest unmedicated supercentenarians studied here, who represent the relatively natural life course of the longest lived humans, may serve as a valuable index group for future studies of supercentenarian, who are more likely to have received potential anti-aging

medications.

Comments

The study was well designed and data have been thoroughly analyzed and clearly reported

Given the world-wide prevalence of older persons of that major increases will occur in the prevalence of every person, it is imperative that we elucidate factors that are associated with high and extreme longevity that go beyond risk factors for diseases that occur more often in older persons.

Aging is the major risk factor for the most deadly clinical diseases and becomes manifested as time-dependent in our organ reserve functions. In other words aging extracts chaos from ordered behavior within our body cells and their interactions with other matrix and with other cells. I would encourage the authors to address even a broader issue in the discussion of their paper. It's aging (time of our existence) itself, not a disease. The effects of time are numerous. Surely, aging may be construed as a slowly progressive disease that inevitably results in death, often without the occurrence of a clinical disease diagnosis. Extreme longevity or early mortality conferred to some on the basis of genetics, life style and environmental effects modulate the effect at which the rate of time-dependent deterioration that we refer to as aging. In this context, a disease can be construed as exaggerated effects of time or accelerated aging.

Response: Thank you for the insightful comments on how our findings on cardiovascular aging and extreme longevity could translate into clinical practice and preventive cardiology in population aging worldwide. We have thoroughly discussed potential roles of cardiovascular and renal aging in extreme longevity in the Discussion, as below.

P10, L7~

“In our study, both NT-proBNP and cystatin C were specifically associated with cardiovascular mortality in the very old, and provided similar prognostic information for mortality beyond the age of 105 years. When these two biomarkers were simultaneously entered into the final models, the association between cystatin C and mortality was attenuated. These results suggest that upregulation of NT-proBNP in

asymptomatic centenarians reflects a compensatory homeostatic response to hemodynamic stress, which arises from the interplay between cardiovascular and, potentially, renal alterations associated with advanced age. Despite its low prevalence (6.4% in those aged 105-109 years and 3.5% in those aged 110 years or older), atrial fibrillation on ECG has a significant prognostic impact in the highest age group, suggesting potential hemodynamic vulnerability in this cohort. “

P13, first paragraph.

“In conclusion, using datasets of the oldest old in Japan, we showed that low levels of circulating NT-proBNP, a potential surrogate for hemodynamic stress, resulting from intrinsic aging in the cardiovascular and renal system, are associated with exceptional survival to the highest ages. These findings identify molecular and pathophysiological pathways that may limit current human longevity. Given the worldwide increase in life expectancy and rising prevalence of older individuals in the total population, understanding the biological effects of aging on major organ systems and the counterregulatory mechanisms associated with high and exceptional longevity has become a public health priority.”

Reviewer #2 (Remarks to the Author):

Associations of cardiovascular biomarkers with exceptional survival to the highest ages. Hirata et al.

This paper describes the investigation of a unique line up of three Japanese cohorts comprising very old people (85-99 years N=531), centenarians (100-104 years N=288), semi-supercentenarians 105-109 years N=572) and supercentenarians (>110 years N=36). The study is focused on the question whether a set of cardiovascular disease markers associate to survival to the highest ages.

To this end, a battery of 9 cardiovascular biomarkers in the circulation were measured in the 1,427 long-lived people and cross-sectional analyses of age strata, as well as prospective analyses on all-cause mortality were performed. The authors concluded that NTproBNP levels in long-lived people of all age groups are associated with and predictive of all-cause mortality. Adding these markers to existing clinical data does not add much to mortality prediction.

Except for the uniqueness of the study population, it is not so clear what exactly the

novelty is in this paper that warrants publication in Nat Commun. NTproBNP is a known predictor of mortality in coronary artery disease, and the current paper adds that this now also seems the case in extremely long-lived people. Given the many data that were so carefully collected in this nice study I would like to urge the authors to highlight much better what the most interesting novelty is of this paper. The paper is presented almost as a search for the best predictive marker that in the end does not predict too much better than classical clinical variable (which is also truly hard to do, most novel markers cannot do that). The paper seems solid in methodology and epidemiological sense. There may be much more to gain in aetiological sense I think.

Response: Thank you for your clear, and thoughtful comments, all of which we found extremely helpful to reconstruct our statistical strategy and discussion. In the revised article, we sought to elucidate aetiological factors underlying relationship between low NT-proBNP levels and extreme survival, and have added statistical analyses. First, I summarize what we have added and changed in Tables and Figures of the revised manuscript, as below:

Item	Changes
Table 1	Added established biomarkers (lipids, HbA1c, CRP, and albumin)
Table 2	Newly constructed to show the best overall set of predictors for mortality, in combining cardiovascular and established biomarkers, as well as clinical and subclinical medical condition using forward stepwise selection. Thus, figure 4 in the original version was deleted.
Table 3	Same as it was in Table 2 in the original version.
Figure 1	Same as it was in the original version.
Figure 2	Added three of established biomarkers (LDL-cholesterol, creatinine, and albumin).
Figure 3	Same as it was in the original version
Figure 4	Newly constructed to show associations between lipids, HbA1c, creatine, CRP, and albumin and all-cause mortality.
Figure 5	Newly added to show association between circulating biomarkers with cardiovascular or non-cardiovascular mortality in the very old cohort aged 85–99 years old.
Figure 6	Added analysis on albumin
Supplementary Fig 1	Same as it was in the original version.
Supplementary Fig 2	Same as it was in the original version.

Supplementary Table 1	Same as it was in the original version.
	Supplementary Table 2 in the original version has been deleted.
Supplemental Table 2	Newly constructed to show correlation between circulating biomarkers with age in female participants only.
Supplementary Table 3	Correlations of albumin with other biomarkers were added.
Supplementary Table 4	Newly constructed to show independent factors associated with levels of NT-proBNP in multivariate linear stepwise regression models.
Supplementary Table 5	Same as Supplementary Table 4 in the original version.
Supplementary Table 6	Newly constructed to show the best overall set of predictors for mortality, in combining cardiovascular and established biomarkers, as well as clinical and subclinical medical condition using forced entry models.
Supplementary Table 7	Same as Supplementary Table 5 in the original version.
Supplementary Table 8	Analysis of albumin were added.

To respond point-by-point to reviewers' comments, we numbered to following comments (#1, #2, #3,,,) .

#1. Some info on the choice of biomarkers and previous work explaining why these were selected (based on previous studies of CVD risk and mortality should be added to the introduction/discussion. The introduction states that CVD risk in centenarians is low so endogenous counter regulation mechanism against CVD are expected to be found among the centenarians. The paper focuses on markers of CVD risk stating that it is known that lipid and IR risk factors are low in the centenarians, although ECG abnormalities have been found. The choice of markers is only explained in experimental procedures. It should be stated in the intro why the markers were selected : indicate heart failure better than other markers ? Especially indicates ECG abnormalities ? The materials and methods by the way explains that four biomarkers were selected, but actually 9 were investigated.

Response: We added several paragraph describing why these nine cardiovascular biomarkers were selected, and how they compare to established biomarkers (HDL LDL, HbA1c, Creatinine, CRP, and albumin) with regard to their utility in mortality prediction. We also added very short summary of our results in the Introduction, which we believe improves overall readability of the manuscript.

P2, L2~

“To test this hypothesis, we selected nine circulating biomarkers reflecting distinct cardioprotective and pathogenic pathways on the basis of previous epidemiological evidence and biological rationale.¹⁷⁻²⁰ Four biomarkers of endogenous cardioprotective molecules include N-terminal pro-B-type natriuretic peptide (NT-proBNP, neurohormonal activity), erythropoietin (erythropoiesis and hypoxic response mediated by hypoxia-inducible factor-1 (HIF1)), adiponectin (insulin-sensitizing and anti-inflammatory pathway), and extracellular superoxide dismutase (EC-SOD or SOD3, antioxidant enzyme in the arterial wall). B-type natriuretic peptide, a bioactive counterpart of NT-proBNP, causes natriuresis and diuresis, arterial dilatation, and antagonism of the renin-angiotensin-aldosterone system, thus counter-regulating hemodynamic abnormalities in heart failure.²¹ All of these cardioprotective biomarkers are upregulated in elderly patients with heart failure.¹⁷⁻²⁰ Three inflammatory mediators include interleukin-6, tumor necrosis factor-alpha (TNF-alpha), and angiotensin-like protein 2 (Angptl2). Angptl2 is upregulated in obesity and type 2 diabetes and accelerates endothelial inflammation, atherosclerosis, and the pathogenesis of heart failure.^{22, 23} Finally, reduced reserve capacity of multiple organ systems is involved in heart failure in old age;²⁴ hence, the levels of two biomarkers, cystatin C and cholinesterase, were measured as indicators of the functional reserves of the kidney and liver, respectively.^{25, 26} Cystatin C was selected because it shows a much higher correlation with age than does creatinine in approximately 5,000 healthy individuals ranging from 25 to 110 years.²⁷ These nine biomarkers were assessed for associations with survival in multiple cohorts of centenarians, (semi)-supercentenarians, and very old individuals, compared with established biomarkers and baseline clinical conditions. First, we show an age-related increase in cardioprotective and inflammatory biomarkers, and a decrease in organ reserves up to 115 years of age. Of these, four biomarkers including NT-proBNP, interleukin-6, cystatin C, and cholinesterase were associated with all-cause mortality in the oldest old. Finally, only the relationship between NT-proBNP and all-cause mortality was robust against adjustment for inflammation, organ reserve, and clinical and subclinical conditions.”

#2. And it was indicated that centenarians are low in lipid and IR risk factors, so one would like to see whether the mortality is predicted better than these, but I could not find lipid or glucose related markers in table 1. Hyperglycaemia and hyperlipidemia are in the base model, but these are the thresholds for diagnosis.

Response: We agree that readers would like to know associations between lipids and glucose metabolism and exceptional survival. Accordingly, we have added these established biomarkers in Table 1, Figure 2, and Figure 4,

#3. The paper is focused on CVD risk but not only on CVD mortality. Is the main cvd mortality death cause heart failure ? Do the chosen markers not indicate risk for other types of death causes ? if death causes are not available then what about other studies ? Is in the four age categories the prevalence of disease and death causes known.

Response: These points are extremely useful for us to consider aetiological factors and biological drivers of mortality at the extremely old ages. First, we have now added Figure 5 to show the relationship between biomarkers and cause-specific mortality in the very old cohort. We found that NT-proBNP and cystatin C are specifically associated with cardiovascular mortality, while albumin and cholinesterase were associated with non-cardiovascular mortality in this cohort. Second, because identifying the cause of death in centenarians can be challenging, particularly in the oldest centenarians such as those studied here, we refer to previous reports including autopsy studies and discussions of cause of death in centenarians in the Discussion, as below.

P10, second paragraph

“Identifying cause of death of centenarians is challenging, because a significant part of this population dies in the non-hospital setting (e.g., nursing home or residential care home) and with asymptomatic clinical presentations. In a population-based study using the death registration of 35,867 centenarians who died in England between 2001 and 2010 (GUIDE Care project), the most prevalent cause of death was “old age” (28.1%), followed by pneumonia (17.7%), cerebrovascular disease (10.0%), and other circulatory diseases (9.8%). Death from ischemic heart disease (8.6%) and cancers (4.4%) are uncommon compared to that among people of younger old age (80–84 years).⁴⁵ Autopsy is the best possible method for accurately identifying the cause of death in the oldest old. In an autopsy study of 40 centenarians who died unexpectedly out of hospital, the most common cause of death was cardiovascular disease (68%), respiratory disease (25%), gastrointestinal disease (5%), and cerebrovascular disease (2%), but no centenarians died from cancer.⁴⁶ Another autopsy study of 140 centenarians and 96 older adults aged 75–95 years found that the prevalence of pathologically confirmed ischemic cardiomyopathies was equivocal (37.8% and 33.3%, in centenarians and older adults,

respectively) but that of acute myocardial infarction was lower (5.9% and 20.5%, $p=0.001$) and that of cardiac amyloidosis was higher (11.3% and 0.0%, $p=0.002$) in centenarians than in older adults.⁴⁷ Given the low cardiometabolic risk in this population, the high NT-proBNP levels of centenarians may reflect cardiac dysfunction attributable to age-related myocardial remodeling.”

#4. The medical conditions at baseline were recorded. Which of these conditions was associated with mortality ? Which of the markers tested was associated with these conditions ? Figure 2 shows association of baseline markers with age for individuals with or without the cardiovascular conditions (three criteria were mentioned: CVD disease history, medication use and ECG abnormalities) . One would like to know which of these predict mortality best (and is that so in each of the age categories) and is that prediction better than that of non CVD medical baseline conditions. And with respect to the markers one would like to know with which relevant medical condition there is baseline association other than CVD conditions.

Response: We agree that intricate associations between clinical and subclinical conditions and biomarkers need to be sorted out. We have added supplementary Table 4 to show independent associations of NT-proBNP concentrations (log-transformed values) with clinical and subclinical conditions and circulating biomarkers. We found age, cystatin C, and atrial fibrillation are strongly associated with NT-proBNP, suggesting that determinants of circulating NT-proBNP in the oldest old are similar to those in the younger population. Second, to examine which of clinical and subclinical conditions or circulating biomarkers best predicts all-cause mortality, these variables are combined in the final model using forward stepwise selection (Table 2). We have added forced entry models with all variables in Supplementary Table 6. The additional statistics confirmed the prognostic importance of NT-proBNP particularly in centenarians and semi-supercentenarians. Interestingly, atrial fibrillation on ECG has significant prognostic impact in the highest age cohort (≥ 105 years), collectively suggesting potential hemodynamic vulnerability at the highest ages. This issue was vigorously discussed in the Discussion as below.

P10, L7~

“In our study, both NT-proBNP and cystatin C were specifically associated with cardiovascular mortality in the very old, and provided similar prognostic information for mortality beyond the age of 105 years. When these two biomarkers were

simultaneously entered into the final models, the association between cystatin C and mortality was attenuated. These results suggest that upregulation of NT-proBNP in asymptomatic centenarians reflects a compensatory homeostatic response to hemodynamic stress, which arises from the interplay between cardiovascular and, potentially, renal alterations associated with advanced age. Despite its low prevalence (6.4% in those aged 105-109 years and 3.5% in those aged 110 years or older), atrial fibrillation on ECG has a significant prognostic impact in the highest age group, suggesting potential hemodynamic vulnerability in this cohort.

#5 A side remark following the discussion: renal function and haemodynamic stress is known to be so relevant in elderly, why is creatinine or the use of diuretics than not among the markers.

Response: Plasma creatinine is included in Table 2, Figure 2, and Figure 4. As compared to cystatin C, creatinine was minimally correlated with age at assessment. Associations between cardiac medications and mortality were not significant in any cohort as shown in Supplementary Table 6,

#6. With respect to Figure 2: The controls as I understand are mainly women (the wives of the oldest son dominating the offspring population). Is the correlation the same if one confines the correlations only to women (the oldest categories mainly consist of women but the very old not).

Response: We have added Supplementary Table 2 to show correlations between biomarkers with age are largely maintained when restricted to female participants only.

#7. The authors in their abstract claim that the findings expand biological knowledge. Burt the paper is focused on testing biomarkers in epidemiological sense (which predicts mortality best).

Please, describe in the discussion not only that there are associations identified, but also in which direction the associations have been found. F.e. in page 8 line215: ..., these results suggest that circulating NT-proBNP is an important biological correlate of exceptional survival to the highest ages => indicate whether this is about low or high levels. Please check the document for the description of direction of effects.

Low NT-proBNP seems to associate with extreme survival: could the authors speculate

in the discussion whether (super)centenarians are reflecting a selection of people with low lifetime risk of cardiovascular disease? Is the working hypothesis now that (super)centenarians have delayed their cardiovascular risk by 30 years? What about other disease and death risks? The discussion section is not so clear yet in the proposed hypothesis.

Response: We have rephrased the “association between NT-proBNP and mortality” to “association between low NT-proBNP levels and survival advantage” in both the abstract and Discussion.

To elucidate potential mortality drivers of the oldest old from fair perspective within our study constructs, we conducted additional statistics as shown in Table 2 and Supplementary Table 6, and Figure 4. These efforts collectively suggest high NT-proBNP, a possible surrogate for cardiovascular and renal aging becomes dominantly correlated with high mortality as they age, and its strongest impacts are observed in the highest age cohort (semi-supercentenarians). Significant prognostic impacts of atrial fibrillation suggest hemodynamic vulnerability of the oldest of the oldest individuals. In contrast, plasma albumin was consistently associated with high mortality across age cohorts, suggesting this biomarker provides less information regarding aetiology of death at advanced ages. Cancers have a less important role on death among centenarians according to literature. This issue is discussed in the Discussion, as below.

P11, last paragraph

“Notably, plasma albumin levels were most consistently associated with all-cause mortality in our oldest old cohort. Albumin is an established biomarker of nutritional status,⁵³ but it also related to inflammation,⁵⁴ hepatic synthesis capacity,⁵⁵ and prognosis of heart failure,⁵⁶ suggesting a multifaceted nature of this biomarker. In this study, plasma albumin levels were significantly correlated with cholinesterase levels, inflammation, and even NT-proBNP levels (age- and sex-adjusted partial correlation coefficients: $r=.409$, $p<0.001$; $r=-.402$, $p<0.001$; $r=-.361$, $p<0.001$; and $r=-.145$, $p<0.001$ for cholinesterase, CRP, interleukin-6, and NT-proBNP, respectively (Supplementary Table 3). Despite overall prognostic utility, plasma albumin is less informative with regard to the etiology of mortality.”

#8. The question is whether in aetiological sense one can show that neurohormonal activation (indicated by NT-proBNP) is more relevant for survival than inflammation,

lipid metabolism, organ reserve, reflected by the other markers.

Response: Our results consistently show that low NT-proBNP is more strongly associated with exceptional survival than inflammation, lipid and glucose metabolism, and organ reserve. The finding may be attributable to the diversity of root causes and pathophysiological roles of inflammation (e.g. atherosclerosis, sarcopenia and frailty, dementia, osteoporosis) in older adults. We address this issue in the revised Discussion as below.

P11, second paragraph

“Inflammation is the hallmark of aging and a cardiovascular pathology⁴⁹; however, its root causes and pathophysiological roles are diverse in older adults.⁵⁰ Our previous study demonstrated that inflammation correlated with physical capability and cognitive function in centenarians and semi-supercentenarians,³⁰ supporting multiple health effects of inflammation at an advanced age. Given a dominance of cardiovascular mortality, pro-inflammatory cytokines may be less prognostic than circulating NT-proBNP at the highest ages.”

#9. Why was only the SOD3 variant investigated.

Response: A series of epidemiological studies has consistently found that a common missense mutation in codon 213 in exon 3 of *SOD3* (rs1799895) is associated with an approximately 10-fold increase in plasma EC-SOD concentrations (ref 32). We have described this point in the Results.

P5, L13~

“A common missense mutation in codon 213 in exon 3 of *SOD3* (rs1799895) has been shown to be associated with an approximately 10-fold increase in plasma EC-SOD concentrations and an elevated risk for incidental ischemic heart disease.³”

#10. If (super)centenarian seem to have a history of low levels of NT-proBNP, it would have been logical to test for the prevalence of SNPs contributing to NT-proBNP levels. rs198389 and rs61761991 in the gene NPPB have previously been associated: are the genotype distributions of these SNPs changing over the age strata? Could one test the causality of the NT-proBNP level to longevity using these SNPs (Mendelian Randomization). There are variants for IL6 (even associated with longevity in previous

studies), adiponectin etc.

Response: We agree that genetic studies such as Mendelian Randomization would be particularly useful to elucidate causal relationship between circulating biomarkers and exceptional survival. We are currently proceeding with whole genome sequencing (WGS) against several hundreds of centenarians. As a result of quantitative trait analysis for NT-proBNP, we isolated a few candidate single nucleotide variations (SNVs) shown in figures below. However, these genetic studies must be done carefully, and it may take a while to replicate our findings in an independent cohort. For these reasons, we'd like to summarize these data in a future report. This issue is described as one of the study limitations in the Discussion.

[redacted]

#11. The authors in their abstract claim that the findings may have public health implications. Which ?

Response: Aging is a dominant risk factor for cardiovascular disease; however, biological overlap between aging and cardiometabolic factors, including body adiposity and hypertension makes it difficult to interpret the extent to which aging itself contributes to cardiovascular outcomes. In the present study, centenarians and (semi-)supercentenarians showed a low prevalence of clinical and subclinical cardiovascular disease detectable by ECG, as well as low cardiometabolic risk profiles. Nevertheless, circulating NT-proBNP continuously increased with age up to 115 years. Our centenarian cohort provides unique opportunity to elucidate clinical relevance of cardiovascular and renal aging in limiting survival chance at the extreme old ages. In the face of heart failure pandemic, driven mainly by population aging, understanding counterregulatory mechanisms that slow cardiovascular aging in supercentenarians may have public priority. Thus, I have added a sentence in the Discussion, as below.

P13, L7

“Given the worldwide increase in life expectancy and rising prevalence of older individuals in the total population, understanding the biological effects of aging on major organ systems and the counterregulatory mechanisms associated with high and

exceptional longevity has become a public health priority.”

Reviewers' comments:

Reviewer #2 (Remarks to the Author):

The Authors have dealt with all comments very carefully. Especially the selection of markers is now more clear. Also additional analyses are now included that increases the value of the paper (such as Figure 5 and Table 2), but also now lead to some confusion, please see the following points:

1. Please check carefully all legends and titles for figures and tables. Title of Fig 2 in the legend is not correct, for example. Also indicate that the HR figures in all figures are based on the base model analysis. Please also explain all notations in tables and Figures if you have not done so (Supp table 4 for example the notation: ...
2. The groups of markers one is investigating in different steps needs carefull explanation, I suggest that authors generate a simple flow chart of analyses including a listing of variates for these analyses so the reader can see which analyses have been done and which marker sets one is talking about in sub analyses. In the basic model one corrects for traditional risk factors. Selected on what basis , factors investigated in the clinic, perhaps, then why are Creatinine or Crp not among traditional risk factors ? Please refer to references if possible. I appreciate that since these data were available, the 'established markers' are now also included in the analysis, but please help the reader to follow the logic of the different steps in analysis and markers that belong to that step. For example Creatinine and CRP are designated as established risk factors. On the basis of what reasoning did one include albumin in the traditional risk factors and not CRP or creatinine, whereas albumin is in both (traditional and established) included.
3. In the first analyses one is investigating the biomarkers for cardioprotection, inflammation and organ reserve. Their mortality prediction is compared to established biomarkers and then in Table 2 and 3 one comes across other combinations of markers that are now called clinical and subclinical (baseline) conditions (how do these relate to the term traditional risk factors or established markers). One has to go to sup table 6 to see which markers were tested (what is meant by clinical, subclinical, medical conditions, to learn that this is a combination of traditional, established and cardioprotect markers and one also observes that markers can also appear just in one age stratum: Atrial fibrillation and old myocardial infarction only in the oldest group (mention this in the text as is, but not in the table to stick to the same variables in all age strata, this table is showing teh tested markers, not only teh effective ones, right ? The paper needs clarity on these issues, so the reader does not have to search in legends and tables of figures and sup tables.

Small issues:

4. Improve legends from table 2 (something like the independent markers resulting from the stepwise selection on traditional, established and candidate markers (positive and negative for cardioprotection).

5. I would change the sentence in the abstract that was added to this version as follows: Of these, only low N-terminal (pro..etc) associated with survival advantage to the supercentenarian stage, independent of traditional cardiovascular risk factors.

6. Please check language of the novel text (s.a. page 6 r 187: remained significantly... instead of remind)

7. In the discussion you mention that the longest lived remain asymptomatic (r245). In your intro you use the argument that cvd is present amongst the longest lived but perhaps cardioprotective factors keep them healthy. INdeed more than half of the longest lived do not use medication, but also 38% has hypertension just as the centenarians, and even more than teh centenarians have hyperlipidemia, so perhaps drugs are not given or not taken.

8. In the discussion p11 r49 you mention that 'Inflammation is the hallmark of aging and a cardiovascular pathology' The papers that use Hallmarks of Ageing mention 9 of them. So please change the sentence into: Inflammation is one of the prominent hallmarks of ageing and also of cardiovascular pathology.

9. I accept indeed that more genetic studies are for teh future, it is good to mention it in the discussion.

Reviewer #3 (Remarks to the Author):

This is a very interesting manuscript. I have some comments for the authors to consider:

1. The whole cohort comprises three different studies put together - in other words, this is a synthetic cohort. The sampling of subjects in different age groups was not considered in the analyses. Clearly, the centenarians were over-sampled compared to oldest-old; and the super-centenarians were over-sampled compared to the other two groups. This may not have a large influence on the identification of prognostic biomarkers. However, this may have a large impact on the actual magnitude of the strength of association between the biomarker and the hazard of mortality. This is an important design issue which needs to be addressed.

2. The abstract seems to have not been revised. It does not describe any results pertaining to albumin, which turns out to be the most powerful predictor of all-cause mortality.

3. I am not sure that the phrase "cardiovascular" is justifiable in the title, since albumin, which is the most powerful predictor is not a cardiovascular marker.

4. The authors state that they used a "multivariable stepwise linear regression" approach. Did they use a "forward" or "backward" selection?

5. Stepwise selection approaches are known to have some important limitations that are well-known in statistical literature (e.g., high variance, biased estimation of regression coefficients). I would suggest that the authors consider a penalized regression approach (e.g., lasso) with 5-fold cross-validation to pick optimal penalty parameter.

ID: NCOMMS 18-33291B

Title: Associations of Cardiovascular Biomarkers with Exceptional Survival to the Highest Ages

Authors: Hirata T, Ara Y, Yuasa S, Abe Y, Takayama M, Sasaki T, Kunitomi A, Inagaki H, Endo M, Morinaga J, Yoshimura K, Adachi T, Oike Y, Takebayashi T, Okano H, and Hirose N.

We appreciate the careful review and constructive suggestions from the two reviewers. First, we added Dr. Kimio Yoshimura to the list of coauthors because of his expertise in suggested statistical analysis such as the LASSO Cox model and bootstrapped analysis. Based on comments from the reviewers and the editorial team, we have addressed design issues and conducted additional analyses, which collectively support the robustness of our results. We are confident that the manuscript has been substantially improved and provides unique insights into the effects of cardiovascular aging on human longevity.

The original reviewer comments are below *in italics*, and our responses appear in regular typeface. Citations from the main text are underlined.

First, we have summarized corrections and modulations in the Tables and Figures of the second revised manuscript, as shown in the table below:

Table: Summary of the Corrections from the First Revision

Item	Changes
Table 1	Added chronic kidney disease (stage 3b-5), established biomarkers are replaced by traditional risk factors (continuous variables).
Table 2	Modified Table 3 in the first revision. (Table 2 in the first revision is now Supplementary Table 6)
Fig 1	Same as it was in the original version.
Fig 2	Same as it was in the first revision.
Fig 3	Newly added to show statistical analysis process.
Fig 4	Same as Fig 3 in the first revision.
Fig 5	Same as Fig 4 in the first revision, but “Established biomarkers” in the first revision were renamed as “Traditional risk factors (continuous variables).”

Fig 6	Newly added to show LASSO-Cox approach to select the best set of prognostic markers.
Fig 7	Same as Fig 5 in the first revision.
Fig 8	Same as Fig 6 in the first revision.
Supplementary Fig 1	Same as it was in the original version.
Supplementary Fig 2	Same as it was in the original version. (The title was corrected.)
Supplementary Fig 3	Newly constructed to show the sensitivity analysis where participants with highest tertile of cystatin C were excluded.
Supplementary Table 1 to Supplementary Table 5	Same as they were in the first version.
Supplementary Table 6	Newly constructed to show the best overall set of predictors for mortality, in combining candidate biomarkers and traditional risk factors using forward stepwise regression. (Modified Table 2 in the first revision.)
Supplementary Table 7	Newly constructed to show the best overall set of predictors for mortality, in combining candidate biomarkers and traditional risk factors using forward stepwise regression. (Modified Supplementary Table 6 in the first revision.)
Supplementary Table 8	Same as Supplementary Table 7 in the first revision.
Supplementary Table 9	Same as Supplementary Table 8 in the first revision

Response to Reviewers' comment and manuscript changes

Reviewer #2 (Remarks to the Author):

The Authors have dealt with all comments very carefully. Especially the selection of markers is now more clear. Also additional analyses are now included that increases the value of the paper (such as Figure 5 and Table 2), but also now lead to some confusion, please see the following points:

- 1. Please check carefully all legends and titles for figures and tables. Title of Fig 2 in the legend is not correct, for example. Also indicate that the HR figures in all figures are based on the base model analysis. Please also explain all notations in tables and Figures if you have not done so (Supp table 4 for example the notation: ...*

Response:

We thank the reviewer for the thorough reading of our manuscript. As was suggested, we have corrected the title of Fig 2 in the legend, and we have revised the footnotes of all figures and tables carefully.

2. The groups of markers one is investigating in different steps needs careful explanation, I suggest that authors generate a simple flow chart of analyses including a listing of variates for these analyses so the reader can see which analyses have been done and which marker sets one is talking about in sub analyses. In the basic model one corrects for traditional risk factors. Selected on what basis, factors investigated in the clinic, perhaps, then why are Creatinine or Crp not among traditional risk factors? Please refer to references if possible. I appreciate that since multivariate models, these data were available, the 'established markers' are now also included in the analysis, but please help the reader to follow the logic of the different steps in analysis and markers that belong to that step. For example Creatinine and CRP are designated as established risk factors. On the basis of what reasoning did one include albumin in the traditional risk factors and not CRP or creatinine, whereas albumin is in both (traditional and established) included.

Response:

First, as suggested, we have created a flowchart describing our statistical procedures with all covariates step by step (Fig 3). Second, established biomarkers (HDL- and LDL-cholesterol, HbA1c, Creatinine, eGFR, and CRP) are now included in the traditional cardiovascular risk factors as continuous variables. Albumin is included in the base model as an independent covariate, because it is not a cardiovascular risk factor, but has a significant prognostic value even in the oldest old. Third, we reconstructed the base model with all previous variables and creatinine (as chronic kidney disease) and CRP as categorical variables. We referred to corresponding references for inclusion of variables into the base model (references 28, 65, 66).

3. In the first analyses one is investigating the biomarkers for cardioprotection, inflammation and organ reserve. Their mortality prediction is compared to established biomarkers and then in Table 2 and 3 one comes across other combinations of markers that are now called clinical and subclinical (baseline) conditions (how do these relate to the term traditional risk factors or established markers). One has to go to sup table 6 to

see which markers were tested (what is meant by clinical, subclinical, medical conditions, to learn that this is a combination of traditional, established and cardioprotect markers and one also observes that markers can also appear just in one age stratum: Atrial fibrillation and old myocardial infarction only in the oldest group (mention this in the text as is, but not in the table to stick to the same variables in all age strata, this table is showing teh tested markers, not only teh effective ones, right ? The paper needs clarity on these issues, so the reader does not have to search in legends and tables of figures and sup tables.

Response:

To make the text and data presentation clearer, we revised our classification of biomarkers and statistical analysis process as shown in Fig 3.

First, we eliminated use of the term “established biomarkers.” Variables that used to be in this category (HDL- and LDL cholesterol, HbA1c, etc.) are now treated as continuous variables of traditional cardiovascular risk factors as shown in Fig 5.

Second, we reconstructed the base model with all previous variables and creatinine (as chronic kidney disease) and CRP as categorical variables (Supplementary Table 5). We referred to corresponding references for inclusion of variables into the base model (references 28, 65, 66).

Third, we revised forced entry models in Supplementary Table 7 (former Supplementary Table 6), with exactly the same covariates included across each age group.

In addition, we have described the classification of comorbidities more carefully in the Method section.

P16, Line 14

“Because of the high percentage of undiagnosed diabetes mellitus in the oldest old population, we defined diabetes as fulfilling one or more criteria: (1) self-reported diagnosis, (2) administration of insulin or other oral hypoglycemic medications, (3) random plasma glucose ≥ 200 mg/dL, or (4) hemoglobin A1c (HbA1c) $\geq 6.5\%$. Hyperlipidemia was defined as a Low-density lipoprotein cholesterol (LDL-C) level ≥ 140 mg/dL, or triglyceride levels ≥ 200 mg/dL, or current use of medication for dyslipidemia. Hypertension was defined as current use of medication for hypertension or self-reported diagnosis.”

Small issues:

4. Improve legends from table 2 (something like the independent markers resulting

from the stepwise selection on traditional, established and candidate markers (positive and negative for cardioprotection).

Response:

We have revised the legend from Supplementary Table 6 (former Table 2) as below:

Independent Prognostic Markers resulting from the Forward Stepwise Selection on Candidate Biomarkers and Traditional Risk Factors in Overall and across Age Groups

5. I would change the sentence in the abstract that was added to this version as follows: Of these, only low N-terminal (pro..etc) associated with survival advantage to the supercentenarian stage, independent of traditional cardiovascular risk factors.

Response:

We have changed the sentence in the Abstract according to both reviewers' suggestion within the word limit:

“Supercentenarians (those aged ≥ 110 years) are approaching the current human longevity limit by preventing or surviving major illness. Identifying specific biomarkers conducive to exceptional survival might provide insights into counter-regulatory mechanisms against aging-related disease. We examined the associations between cardiovascular disease-related biomarkers and survival to the highest ages using a unique dataset of 1,427 oldest individuals from three longitudinal cohort studies, including 36 supercentenarians, 572 semi-supercentenarians (105-109 years), 288 centenarians (100-104 years), and 531 very old people (85-99 years). During follow-up, 1,000 participants (70.1%) died. Overall, N-terminal pro-B-type natriuretic peptide (NT-proBN), interleukin-6, cystatin C and cholinesterase were associated with all-cause mortality independent of traditional cardiovascular risk factors and plasma albumin. Of these, low NT-proBNP levels were statistically associated with a survival advantage to supercentenarian age. Only low albumin was associated with high mortality across age groups. These findings expand our knowledge on the biology of human longevity.” (150 words)

6. Please check language of the novel text (s.a. page 6 r 187: remained significantly... instead of remind)

Response:

The manuscript has undergone English editing by Editage, with particular attention to the novel text.

7. In the discussion you mention that the longest lived remain asymptomatic (r245). In your intro you use the argument that cvd is present amongst the longest lived but perhaps cardioprotective factors keep them healthy. INdeed more than half of the longest lived do not use medication, but also 38% has hypertension just as the centenarians, and even more than teh centenarians have hyperlipidemia, so perhaps drugs are not given or not taken.

Response:

We agree with the possibility that the oldest centenarians might be under prescribed anti-hypertensive drugs or other cardiovascular medications because of underlying frailty and cognitive decline. A higher percentage of supercentenarians free from cardiovascular medication use does not necessarily mean that they are free from cardiovascular illness. So, we changed description as below:

Page 9, Line 8:

“Despite these indicators, the longest-lived individuals exhibited a relative absence of overt cardiovascular disease and 56.3% of the supercentenarians in the present study were not taking cardiovascular medications.”

We also added some sentences in the study limitation section as below;

Page 13, Line 5:

“Third, we obtained participants’ medication lists at baseline only, which may not represent life-time exposure to certain medications. In clinical practice, physicians are not without the option of deprescribing anti-hypertensive drugs as their patients get older and frailer.⁶⁰ This may at least partly mediate a paradoxical association between hypertension and better survival in the oldest old⁶¹ and even in our own centenarian cohort (Fig 6e). Therefore, we limited cardiovascular medication to four classes of drug in survival analyses: nitrate, oral anticoagulant, antiarrhythmic drug and digoxin, which are less likely to be withdrawn even in the extremely old.”

8. In the discussion p11 r49 you mention that 'Inflammation is the hallmark of aging and a cardiovascular pathology' The papers that use Hallmarks of Ageing mention 9 of

them. So please change the sentence into: Inflammation is one of the prominent hallmarks of ageing and also of cardiovascular pathology.

Response:

We have changed the sentence according to the reviewer's suggestion and cited a corresponding reference.

Page 12, second paragraph:

"Inflammation is one of the prominent hallmarks of aging⁵⁰ and also of cardiovascular pathology;⁵¹"

9. I accept indeed that more genetic studies are for teh future, it is good to mention it in the discussion.

Thank you for your kind comments to our ongoing genetic study.

Reviewer #3 (Remarks to the Author):

This is a very interesting manuscript. I have some comments for the authors to consider:

1. The whole cohort comprises three different studies put together - in other words, this is a synthetic cohort. The sampling of subjects in different age groups was not considered in the analyses. Clearly, the centenarians were over-sampled compared to oldest-old; and the super-centenarians were over-sampled compared to the other two groups. This may not have a large influence on the identification of prognostic biomarkers. However, this may have a large impact on the actual magnitude of the strength of association between the biomarker and the hazard of mortality. This is an important design issue which needs to be addressed.

Response:

We thank the reviewer for pointing out this important design issue. As we agree with the importance of this issue, and we have addressed it in study limitations as below:

Page 12, last paragraph:

“Second, there is some heterogeneity between the original three studies from which the entire cohort was aggregated. Obviously, semi-supercentenarians were oversampled, because the prime aim of this cohort was to discover the biological and genetic basis of supercentenarians an extraordinarily rare phenotype. Although the effect sizes of each biomarker on mortality were generally similar across the age-stratified cohorts, there were differences in directions of effects of some biomarkers (i.e. adiponectin and LDL cholesterol). Therefore, the results of such biomarkers in the total combined cohort should be cautiously interpreted.

2. The abstract seems to have not been revised. It does not describe any results pertaining to albumin, which turns out to be the most powerful predictor of all-cause mortality.

Response:

We have revised the Abstract as follows:

“Supercentenarians (those aged ≥ 110 years) are approaching the current human longevity limit by preventing or surviving major illness. Identifying specific biomarkers conducive to exceptional survival might provide insights into counter-regulatory mechanisms against aging-related disease. We examined the associations between cardiovascular disease-related biomarkers and survival to the highest ages using a unique dataset of 1,427 oldest individuals from three longitudinal cohort studies, including 36 supercentenarians, 572 semi-supercentenarians (105–109 years), 288 centenarians (100–104 years), and 531 very old people (85–99 years). During follow-up, 1,000 participants (70.1%) died. Overall, N-terminal pro-B-type natriuretic peptide (NT-proBN), interleukin-6, cystatin C and cholinesterase were associated with all-cause mortality independent of traditional cardiovascular risk factors and plasma albumin. Of these, low NT-proBNP levels were statistically associated with a survival advantage to supercentenarian age. Only low albumin was associated with high mortality across age groups. These findings expand our knowledge on the biology of human longevity.” (150 words).

3. I am not sure that the phrase "cardiovascular" is justifiable in the title, since albumin, which is the most powerful predictor is not a cardiovascular marker.

Response:

We agree that albumin is the most powerful predictor, but not a cardiovascular risk factor, and we realize that this distinction is very important. Accordingly, we treated albumin as an independent predictor from cardiovascular markers in the text and all analyses. Furthermore, we have added several sentences (details below) to carefully explain the distinctive predictive roles between albumin and NT-proBNP regarding age-specific mortality. Although we agree that albumin is the strongest predictor across the studied age spectrums, our prime aim of this study was to address the biological mechanisms that confer exceptional survival to the highest ages, particularly in view of cardiovascular protection. In this sense, plasma albumin is less informative as an etiological mortality driver, so we have respectfully opted to keep the original title of our article.

Page 11, Line 15:

“These results suggest that upregulation of NT-proBNP in asymptomatic centenarians reflects a compensatory homeostatic response to hemodynamic stress, which arises from the interplay between cardiovascular and, potentially, renal alterations associated with advanced age and ultimately limiting chances of survival to the supercentenarian age.”

Page 13, Line 2:

“In contrast to age-related increase in prognostic relevance of NT-proBNP, the strength of associations between low albumin and high mortality is markedly stable across age groups.”

Page 13, Line 12

“Collectively, low albumin levels in the oldest old may represent common debilitating processes across a spectrum of ages and pathophysiologies.”

4. The authors state that they used a "multivariable stepwise linear regression" approach. Did they use a "forward" or "backward" selection?

Response:

We have corrected the term as “multivariable stepwise linear regression with backward elimination.” Page 5, Last paragraph, and Page 19, Line 1

5. Stepwise selection approaches are known to have some important limitations that are well-known in statistical literature (e.g., high variance, biased estimation of regression

coefficients). I would suggest that the authors consider a penalized regression approach (e.g., lasso) with 5-fold cross-validation to pick optimal penalty parameter.

Response:

We thank the reviewer for the suggestion to apply a penalized regression approach. Accordingly, we compared three statistical models, namely LASSO-Cox, forward stepwise regression, and forced entry model to select the best overall set of predictors from the candidate biomarkers and traditional risk factors. We showed the results of LASSO Cox in Fig 6, and those of stepwise regression and forced entry in Supplemental Table 6 and Supplemental Table 7, respectively. We found consistent associations of NT-proBNP with mortality in centenarians (100-104, and 105+), associations of albumin with mortality in all age groups across all three models, confirming the robustness of our results.

Page 19, Line 18:

“Finally, to identify the best overall set of predictors, all the prognostic biomarkers significantly associated with mortality in the multivariate analysis were combined with clinical covariates in the base model. We employed three different techniques to select the most useful prognostic markers in the final model: 1) the least absolute shrinkage and selection operator for Cox regression (LASSO-Cox) with five-fold cross-validation, 2) a stepwise forward selection with inclusion criteria $p < 0.20$, and 3) forced entry models. The LASSO is a penalized technique for variable selection that is effective when the number of events per variables is low.^{67, 68} To standardize the number of participants for the multiple biomarker-risk factor comparisons, we restricted subsequent analysis to participants with complete data on all biomarkers being studied for all-cause mortality.”

Editorial comments

In addition, the editorial team has concerns regarding the C statistics presented in Table 3 and after consultation with Reviewers 2 and 3 we would kindly ask you to please employ additional approaches to ensure these are correct. Firstly, we would suggest that you perform bootstrapped analysis (e.g., with 100 bootstrapped samples) of all of the models and calculate the averaged C-statistics. Secondly, we would suggest that you consider adjusting for any over-estimate of the truth by using, for example, one of the approaches describe here: <https://www.ncbi.nlm.nih.gov/pmc/articles/PMC4108045>. Please highlight all changes in the manuscript text file.

We thank the editorial team for this important suggestion. We have conducted a bootstrap analysis for calculating C-statistics for all the models with 2,000 resampling, by using *pROC* package as shown in Table 2. The results were confirmed by the same analyses described in the paper (Smith GC, et al. Am J Epidemiol. 180, 318-324. doi: 10.1093/aje/kwu140).

Page 19, Line 31;

“To correct optimal prediction in relatively small data sets, empirical 95% CIs and P-values were calculated using a bootstrap approach with 2,000 resampling. ⁶⁹”

Page 20, last paragraph;

Analyses were performed using STATA SE 13 software (Stata Corp LP, College Station, TX, USA), and R (version 3.4.3. for LASSO selection, and version 3.5.1. and *pROC* package for bootstrapped analyses).

Reviewers' comments:

Reviewer #2 (Remarks to the Author):

I think the authors have dealt with all the comments very well, the paper has seriously improved in clarity and some very relevant analytic additions.

Reviewer #3 (Remarks to the Author):

1. I mentioned about the cardiovascular risk factors in the title as being misleading, since Albumin is the single most consistent and strongest risk factor across all 3 age groups and it is not a cardiovascular factor. Hence the title is not justified in my opinion.

2. The authors say that they used the methods in Smith et al, which I had recommended, for correcting the optimism of c-statistics. However, I do not see the corrected c-statistics. If they had actually employed the bootstrap method correctly, the c-statistics should be substantially smaller.

3. The authors claim to have used the lasso approach and found the same results as stepwise selection. While I can see the two approaches selecting the same biomarkers in the model, the coefficients are likely to differ due to penalization. How did they cross-validate the penalty for lasso model? Can they show the results comparing stepwise approach to lasso approach in a supplemental table?

ID: NCOMMS 18-33291C

Title: Associations of Cardiovascular Biomarkers and Plasma Albumin with Exceptional Survival to the Highest Ages

Authors: Hirata T, Arai Y, Yuasa S, Abe Y, Takayama M, Sasaki T, Kunitomi A, Inagaki H, Endo M, Morinaga J, Yoshimura K, Adachi T, Oike Y, Takebayashi T, Okano H, and Hirose N.

We appreciate the careful review and instructive statistical suggestions from the reviewers and editor. We have addressed all issues raised by reviewer 3 and added a new supplementary table (Supplementary table 6) to show the LASSO coefficients converted to hazard ratio. During the revision process, we have updated affiliation of Dr. Takumi Hirata, and funding information from Keio University according to current situation. We also have corrected errors in typography and formatting in references, all of which are underlined. We are confident that the manuscript has been significantly improved and provides unique insights into the effects of cardiovascular aging on human longevity.

The original reviewer comments are below *in italics*, and our responses appear in regular typeface. Citations from the main text are underlined.

Response to Reviewers' comment and manuscript changes

Reviewer #3 (Remarks to the Author):

1. I mentioned about the cardiovascular risk factors in the title as being misleading, since Albumin is the single most consistent and strongest risk factor across all 3 age groups and it is not a cardiovascular factor. Hence the title is not justified in my opinion.

Response

We have changed our title as "Associations of Cardiovascular Biomarkers and Plasma Albumin with Exceptional Survival to the Highest Ages".

2. The authors say that they used the methods in Smith et al, which I had recommended, for correcting the optimism of c-statistics. However, I do not see the corrected

c-statistics. If they had actually employed the bootstrap method correctly, the c-statistics should be substantially smaller.

We thank the reviewer for this detailed suggestion. We now see that our description on C statistics and the optimism in the previous version was insufficient, and we have redone the analysis. First, we found that our prediction models became unstable for centenarians (100–104 years) and for semi-supercentenarians (105 years), when we excluded those who were censored within five years. This may be due in part to the very low 5-year survival rate (14.2% and 1.8% for those aged 100–104 years at baseline and those aged 105 years or older, respectively). Therefore, C statistics was calculated for one-year mortality for those aged 100 years and older. Second, we estimated the optimism of the developed models following Harrell et al (ref 3 in Smith et al), where the estimated optimism was calculated as naïve C-statistics for the bootstrapped samples minus C-statistic evaluated on the original data set. The process was repeated 100,000 times and averaged to gain an accurate estimate of the optimism to the third decimal place. Accordingly, we modulated Table 2 and description on statistical analysis in the main text as below;

P20, L5 (statistical analysis)

The prognostic discrimination for 5-year all-cause mortality of each biomarker was assessed by comparing the base model with the models that incorporated each biomarker with the base model using Harrell's C statistic. According to very low survival rate at five years (14.2% and 1.8% for those aged 100-104 years at baseline and those aged 105 years or older, respectively), C statistics was calculated for one-year mortality for those aged 100 years and older. To correct the overfitting of the statistical models in relatively small data set,⁶⁹ we estimated the optimism of the developed models according to Harrell et al,⁷⁰ where the estimated optimism was calculated as naïve C-statistics for the bootstrapped samples minus C-statistic evaluated on the original data set. This process was repeated 100,000 times and averaged to gain an accurate estimate of the optimism to the third decimal place. The optimism was then subtracted from the C statistics of the original model to provide optimism-corrected C statistics.

P8, first paragraph (results)

In the entire cohort, addition of NT-pro BNP or cholinesterase to the base model marginally or significantly improved discrimination for 5-year all-cause mortality with

small optimism (NT-proBNP: C-statistic, 0.894 [95% CI; 0.871-0.917], p=0.057, optimism-corrected C statistics=0.882; cholinesterase: C-statistic, 0.896 [95% CI; 0.874-0.919], P=0.016, optimism-corrected C statistics=0.885, respectively, Table 2). When stratified by age, the predictivity of the models substantially declined, suggesting that age itself is a dominant prognostic factor. The base model is only weakly predictive of one-year mortality beyond 105 years of age (C-statistic, 0.638 [95% CI; 0.569–708], optimism-corrected C statistics=0.561). Despite wider optimism, adding set of biomarkers significantly and modestly improve the predictivity at the highest ages (all biomarkers: C-statistic, 0.702 [95% CI; 0.639-765], P=0.017, optimism-corrected C statistics=0.623, 0.062 increment from the base model).

Table 2, title

The Prognostic performances for 5-year or 1-year all-cause mortality in the entire and age-stratified cohort

3. The authors claim to have used the lasso approach and found the same results as stepwise selection. While I can see the two approaches selecting the same biomarkers in the model, the coefficients are likely to differ due to penalization. How did they cross-validate the penalty for lasso model? Can they show the results comparing stepwise approach to lasso approach in a supplemental table?

We have added Supplementary Table 6 to show the LASSO coefficients, which are converted to hazard ratios as convenient for comparison with the results from the corresponding stepwise analysis. The original Supplementary Table 6 and subsequent supplementary items have been renumbered Supplementary Table 7, etc.. In addition, we have described the details of LASSO approach in statistical analysis section as below;

P19, last sentence

LASSO shrinks coefficients for weaker predictors toward zero. The degree of shrinkage is determined by an optimal parameter lambda, as identified by five-fold cross-validation.

P20, last paragraph

Analyses were performed using STATA SE 13 software (Stata Corp LP, College Station, TX, USA), and R (version 3.4.3. and glmnet package for LASSO selection; version 3.5.1. and pROC package for calculating C statistics and optimisms). All P values were two-tailed, and a P value < 0.05 was considered statistically significant.

Fig 6 legend

LASSO coefficient profiles of 17 markers associated with mortality were generated for the entire sample (a), those aged 85-99 years at enrollment (b), 100–104 years at enrollment (c), and 105 years or older at enrollment (d). Vertical lines were drawn at the optimal values by using five-fold cross validation. Lasso coefficients of 17 markers are shown in Supplementary Table 6.

Supplementary Table 6, footnote,

LASSO shrinks coefficients for weaker predictors toward zero (denoted as ...). The degree of shrinkage is determined by an optimal parameter lambda.mins, the value of lambda that gives minimum mean cross-validated error are 0.00947, 0.03622, 0.12483, and 0.10931 for the entire cohort (a), those aged 85-99 years at enrollment (b), 100-104 years (c), and 105 years or older (d). LASSO coefficients are converted to hazard ratios as convenient for comparison with the results from the stepwise analysis (Supplementary Table 7).

Reviewers' comments:

Reviewer #3 (Remarks to the Author):

Thank you for addressing all of my comments. The statistical methods are now vastly improved. I still have one major question.

In Table 2 results, the sample sizes are greatly decreased from the original sample size. For example, the entire cohort has $N=1,427$, but in Table 2 the N is only 750. I thought this could be due to missing biomarkers. But, all biomarkers have at least 1,300 measurements, with the exception of NT-proBNP, which has $N=1,080$. Even if you account for dropping those who were censored ($N=86$), the sample size should be much larger.

Can you please explain this?

An additional comment is that strictly speaking it is not appropriate to delete those who are censored. You should use methods for calculating c-statistics which account for censoring. Please check the literature for this.

ID: NCOMMS 18-33291D

Title: Associations of Cardiovascular Biomarkers and Plasma Albumin with Exceptional Survival to the Highest Ages

Authors: Hirata T, Arai Y, Yuasa S, Abe Y, Takayama M, Sasaki T, Kunitomi A, Inagaki H, Endo M, Morinaga J, Yoshimura K, Adachi T, Oike Y, Takebayashi T, Okano H, and Hirose N.

We appreciate the feedback and suggestions from the reviewers and editor. We have addressed an important statistical issue raised by reviewer 3 and revised Table 2 in the main text. We also have corrected a few typographical errors, all of which are underlined. We are confident that the revisions have significantly improved the quality of the manuscript, which provides unique insights into the effects of cardiovascular aging on human longevity.

The original reviewer comments are below *in italics*, and our responses appear in regular typeface. Citations from the main text are underlined.

Response to Reviewers' comment and manuscript changes

Reviewer #3 (Remarks to the Author):

In Table 2 results, the sample sizes are greatly decreased from the original sample size. For example, the entire cohort has $N=1,427$, but in Table 2 the N is only 750. I thought this could be due to missing biomarkers. But, all biomarkers have at least 1,300 measurements, with the exception of NT-proBNP, which has $N=1,080$. Even if you account for dropping those who were censored ($N=86$), the sample size should be much larger.

Can you please explain this?

Response

In addition to missing blood biomarkers and censored survival data, several cases had missing ECG ($N=1,196$) and education status ($N=1,362$) data, which are independently obtained during data collection; this collectively reduced the sample sizes for multiple testing.

An additional comment is that strictly speaking it is not appropriate to delete those who are censored. You should use methods for calculating c-statistics which account for censoring. Please check the literature for this.

We thank the reviewer for this insightful advice on statistical analysis. We agree not to delete censored survival data from our prediction models in Table 2, but to use more sophisticated method to accommodate censored information. We computed Concordance index based on Cox proportional hazard models to assess the prognostic performances of biomarkers. Accordingly, we have revised Table 2 and have modified statistical analysis in the main text as follows:

P8, first paragraph,

In order to assess the prognostic performance of each biomarker, we computed Concordance index (C-index) based on Cox proportional hazard models.⁶⁹ In the entire cohort, the addition of each prognostic biomarker in Fig. 4 (NT-proBNP, interleukin-6, cystatin C, and cholinesterase) to the base model significantly improved risk prediction with relatively small optimism (Table 2). When stratified by age, the predictivity of the models substantially declined, suggesting that age itself is a dominant prognostic factor. The base model is only weakly predictive of mortality beyond 105 years of age (C-index, 0.617 [95% CI; 0.577–0.656], optimism-corrected C-index=0.588). Nevertheless, adding NT-proBNP and, to a lesser degree, cholinesterase significantly improve the predictivity at the highest ages (NT-proBNP: C-index, 0.653 [95% CI; 0.615–0.691], P=0.001, optimism-corrected C-index=0.625; Cholinesterase: C-index, 0.636 [95% CI; 0.596–0.676], P=0.019, optimism-corrected C-index=0.609, respectively, Table 2).

P20, second paragraph,

To assess the prognostic performance of each biomarker, we computed Concordance index (C-index); we compared the base model with the models that incorporated each biomarker with the base model by calculating C-index based on Cox proportional hazard models.⁶⁹ P values for the C-index are computed by assuming asymptotic normality.

P20, last paragraph

Analyses were performed using STATA SE 13 software (Stata Corp LP, College Station, TX, USA), and R (version 3.4.3. and glmnet package for LASSO selection; version 3.5.1.

and survcomp package for calculating C-index and optimisms). All P values were two-tailed except for the C statistics based on Cox hazard models, and a P value < 0.05 was considered statistically significant.

Revised title of Table 2. The Prognostic performance for all-cause mortality in the entire and age-stratified cohort based on Cox proportional hazard models

To the editor,

In addition to above modifications, we have removed one sentence in Results section, because it did not necessarily represent the contents of Table 1.

P5, L7

The prescription rates for cardiovascular medications other than diuretics were lower in centenarians than in the very old (deleted).